# Multi-ancestry investigation of the genomics of erectile dysfunction

Uri Bright[1], Yu Chen [1], Joseph D. Deak [1,2], Hang Zhou [1,2], Daniel F. Levey [1,2] & Joel Gelernter [1,2,3] ✉

Erectile dysfunction is attributable to numerous biological and psychological issues, and its prevalence increases with age. We conducted genome-wide association studies of erectile dysfunction in AllofUs subjects of European and African ancestry, then meta-analyzed our findings with published datasets [$N_{European}$ = 913,194 (136,867 cases); $N_{African}$ = 125,315 (51,599 cases)]. We identified 40 independent variants in Europeans, two in Africans, and 51 cross-ancestry. In all analyses, the strongest effect variants mapped to a non-coding region known to regulate *SIM1*, previously associated with erectile dysfunction: rs78677597 (Europeans) ($p = 5.32 \times 10^{-139}$), and rs17185536 (Africans ($p = 1.17 \times 10^{-9}$) and cross-ancestry ($p = 5.3 \times 10^{-138}$)). Genetic correlations with psychiatric and health traits were moderate. Positive associations with phenotypes related to sexual drive may reflect ascertainment bias. This study is consistent with indications that erectile dysfunction is a complex trait influenced by multiple factors. Our findings emphasize the need to investigate genetic risk – *SIM1* in particular further – to understand the mechanism through which they affect erectile function.

Erectile dysfunction (ED) is the inability of men to gain and maintain an erection in a sufficient manner for sexual performance. The prevalence of ED increases with age: an average of 30% of men over the age of 40 report ED[1], compared to <10% in people younger than 40[2]. Penile erection is usually gained by smooth muscle relaxation, a result of decreased noradrenaline (NA) and increased nitric oxide (NO) release, which leads to enhanced concentrations of cyclic GMP (cGMP), enabling blood flow into the lacunar spaces in the erectile tissues[3]. The main treatment for ED is with phosphodiesterase type 5 (PDE5) inhibitors, which block cGMP degradation by PDE5, thus increasing cGMP-mediated smooth muscle relaxation, facilitating penile erection[4].

The most common direct cause of organic ED is reduced blood flow and arterial insufficiency, which may result from vascular disease, often in association with smoking and diabetes. Besides age, diabetes is the strongest risk factor for ED, and cardiovascular diseases are strongly implicated too[5,6]. There is also high comorbidity between obesity and ED: up to 80% of the people who report ED are overweight or obese[7]. This association between cardiovascular traits, obesity, and ED is strengthened by findings of improved sexual functioning due to lifestyle changes aimed at lowering cardiovascular risk[8]. Other factors include neuronal signaling deficits due to an injury or disease, and hormonal changes, mainly low testosterone[3]. Psychological factors can also play a major role in psychogenic ED; that is, ED that is predominantly caused by psychological factors such as symptoms of anxiety and stress without an organic explanation[9]. In addition, individuals with depression have a 39% higher odds of experiencing ED (pooled OR = 1.39)[10], and men with ED are about twice as likely to have depression[10,11]; there is also a 77% increased risk of anxiety in men with ED[11]. Among people with post-traumatic stress disorder (PTSD), ED is five times more likely to occur than in the general population[12]. ED is also correlated with changes in brain morphology, including decreased postcentral gyrus and increased temporal gyrus volumes, and with altered connectivity between several cortical regions[13,14]. Substance use, such as high alcohol consumption[15], cigarette smoking[16], and cannabis use[17] are risk factors for ED as well. Physical

[1]Department of Psychiatry, Yale School of Medicine, New Haven, CT, USA. [2]Veterans Affairs Connecticut Healthcare System, West Haven, CT, USA. [3]Departments of Genetics and Neuroscience, Yale School of Medicine, New Haven, CT, USA. ✉e-mail: joel.gelernter@yale.edu

activity, on the other hand, reduces the risk of ED and, in some cases, is used as a treatment for ED[18].

A twin study showed that ED is genetically influenced[19], which was confirmed by three genome-wide association studies (GWAS) of ED[20–22]. All three studies showed a significant effect of single-nucleotide polymorphisms (SNPs) mapping to a non-coding region on chromosome 6, in an eQTL region that downregulates the expression of the single-minded 1 (*SIM1*) transcription factor. The lead variants identified in these studies are rs17185536[20], rs57989773[21], and rs78677597[22], all residing in the same LD region, spanning 19 kb. Deficiency of the SIM1 protein is mostly associated with hyperphagia, weight gain, and obesity, seen both in animal studies[23–25] and in humans[26–28].

PDE5 inhibitors, the most common treatment for ED, are considered fairly safe, and their success rate is 65–70%[29]. While these are relatively high percentages, it still means that a significant number of individuals do not respond to the treatment, highlighting the importance of studying this phenomenon further for a better understanding of its biology and the potential for pointing towards novel pharmacotherapies.

In this study, we aimed to investigate the genetic architecture of ED. Our analysis was based on electronic health record (EHR) information, mostly based on self-report (albeit to a physician), and hence likely to be missing a portion of the population who do not report ED, even though they experience ED symptoms[30,31]. Therefore, we defined our phenotype as EHR-ED to differentiate our phenotype from population-based ED figures. EHR-ED patients are likely to be seeking treatment; people with ED who do not seek treatment, e.g., because they are not or do not wish to be sexually active or cannot overcome embarrassment, are less likely to receive an EHR diagnosis. We conducted a GWAS of EHR-ED in subjects of European (EUR) and African (AFR) ancestries using the All of Us (AoU) biobank, then meta-analyzed our results with previously published ED GWAS data, which include a total of 824,472 subjects of EUR ancestry and 94,867 individuals of AFR ancestry[21,22,32]. We then used several post-GWAS techniques, including gene-based analysis, global and local genetic correlation analyses, transcriptome-wide association study (TWAS), phenome-wide association study (pheWAS), polygenic risk scores (PRS) analysis, and genomic structural equation modeling (gSEM) to investigate the genetic factors that underly ED, and their association with other traits, including substance use, psychiatric disorders, personality, lifestyle, and general health.

## Results

### Genome-Wide Association Study (GWAS)
88,722 EUR subjects (16,983 cases) and 30,448 AFR subjects (4215 cases) from AoU (version 8) were included in this study. Cases were on average 10 years older than controls; prevalence of prostate cancer was 3.3 times higher in ED EUR subjects (compared to controls) and 8.4 times higher in AFR; type 2 diabetes (T2D) prevalence was twice as high in ED EUR subjects and four times higher in AFR (Supplementary Table 1). In EUR, we identified one GWS locus—rs17185536 ($p = 1.09 \times 10^{-12}$)—located in a non-coding region on chromosome 6 (Supplementary Fig. 1). In AFR, we did not find any significant SNPs. We performed meta-analyses in EUR and AFR, combining our findings from AoU with previously published data of ED from various sources[21,22,32]. In total, we assembled a sample size of 913,194 EUR (136,867 cases) and 125,315 AFR (51,599 cases) participants (Table 1). We used FUMA to identify lead SNPs and independent genomic risk loci. In the EUR meta-analysis, we identified a total of 40 lead SNPs in 27 genomic risk loci; nine are novel, and for 17 the minor allele is protective against the trait (full details regarding chromosome, position, and $Z$ scores are presented in Supplementary Data 1). The strongest associations were with rs78677597 ($p = 5.35 \times 10^{-139}$), located within the non-coding RNA

gene *LOC105377911*, followed by rs8141413 (intronic to *PHF21B*; $p = 4.18 \times 10^{-19}$), rs138042437 (intronic to *CASC19*; $p = 7.7 \times 10^{-17}$), rs2347923 (intronic to *ESR1*; $p = 2.42 \times 10^{-14}$), and rs10928225 (intronic to *TEX41*; $p = 6.84 \times 10^{-14}$) (Fig. 1a). In AFR, we found two statistically significant variants: rs17185536-T ($p = 1.17 \times 10^{-9}$) and rs55659406 (intronic to *RABGAP1L*; $p = 4.79 \times 10^{-8}$), the latter is novel and for both the minor allele increases risk of EHR-ED (Fig. 1b, Supplementary Data 1). Cross-ancestry (EUR-AFR) meta-analysis revealed an increase from 40 lead SNPs in EUR and 2 in AFR to 51 lead SNPs (34 genomic risk loci, for 24 the minor alleles are protective), with rs17185536 the most significant one ($p = 5.30 \times 10^{-138}$), followed by rs8141413 ($p = 2.63 \times 10^{-21}$), rs138042437 ($p = 7.70 \times 10^{-17}$), rs6557171 (intronic to *ESR1*; $p = 2.79 \times 10^{-15}$), and rs10200647 (intronic to *TEX41*; $p = 2.95 \times 10^{-14}$) (Fig. 1c, Supplementary Data 1). Quantile–quantile (QQ) plots of the EUR, AFR, and cross-ancestry meta-analyses are presented in Supplementary Fig. 2. For improved visibility of the significant SNPs apart from the strongest findings, truncated Manhattan plots of the EUR and cross-ancestry meta-analyses are presented in Supplementary Figs. 3 and 4.

### SNP-based heritability and Inter-Cohort genetic correlations
We used LDSC to calculate the liability-scaled heritability estimates ($h^2$) of all the individual EUR cohorts of ED (Supplementary Table 2) and inter-cohort genetic correlations ($r_g$). These correlations ranged from $r_g = 0.48–1.017$, besides two cases of non-significance, both involving a dataset with a limited sample size (see ref. 21) (Supplementary Table 3). SNP-based heritability for the meta-analyzed data was $h^2 = 0.062 \pm 0.004$.

### Polygenic Risk Scores (PRS) analysis
After Bonferroni correction ($p$-value threshold = 0.0023), in EUR we found that EHR-ED (meta-analysis excluding AoU) PRS is significantly associated with EHR-ED in the AoU sample, in all but two of the 11 examined SNP inclusion $p$-value thresholds (the ones without significant association were $p = 0.01$ and $p = 0.0001$). The strongest effect was achieved using a SNP inclusion cutoff of $p = 0.5$, explaining 9.2% of the phenotypic variance ($p = 2.20 \times 10^{-36}$) (Supplementary Table 4). Nevertheless, the predictive value of this model was limited (AUC = 0.52), in accordance with positive modest association between EHR-ED PRS (under a p-value threshold of 0.5) and EHR-ED risk (Supplementary Fig. 5). Individuals in the top 2.5%, 5% and 10% of predicted EHR-ED probability had 53% (OR = 1.53 ± 0.14), 18% (OR = 1.18 ± 0.13) and 16% (OR = 1.16 ± 0.08) higher odds of EHR-ED compared to individuals in the middle 50% of predicted probabilities. ROC analysis identified an optimal probability cutoff for sensitivity and specificity for SNP inclusion in PRS at $p = 9.23 \times 10^{-6}$. Using this threshold, 4697 cases were correctly predicted (true positives) and 17,583 controls were incorrectly predicted as high risk (false positives), yielding a sensitivity of 0.28 and specificity of 0.75 (Supplementary Table 5).

In the AFR samples, we found no significant effects in any of the examined $p$-value thresholds, although all but one were nominally significant ($p < 0.05$). The strongest association was achieved using a cutoff of $p = 0.2$: $r_{Nagelkerke}^2 = 0.128$ (estimate = 1214, $p = 0.0037$) (Supplementary Table 4).

### MAGMA gene-based and gene set analyses
Using MAGMA gene-based analysis, we found 21 significant genes associated with EHR-ED in EUR. The strongest effects were of *PHF21B* ($p = 1.24 \times 10^{-10}$), *NTNG1* ($p = 5 \times 10^{-10}$), and *CTNNB1* ($p = 5 \times 10^{-10}$) (Supplementary Data 2). In AFR, we found one gene—*RC3H1* ($p = 7.31 \times 10^{-7}$)—that was associated with EHR-ED (Supplementary Data 3). In the cross-ancestry, we found 23 genes associated with ED. The strongest effects were at *ESR1* ($p = 1.98 \times 10^{-11}$), *TBX20* ($p = 5 \times 10^{-10}$), and *ANGPT2* ($p = 6.08 \times 10^{-10}$) (Supplementary Data 4).

**Table 1 | Sample size across the various cohorts**

| Ancestry | Cohort | Source[a] | Phenotype | Cases | Controls | Total | Effective[b] |
|---|---|---|---|---|---|---|---|
| EUR | AoU | | ICD-10, drug prescription record | 16,983 | 71,739 | 88,722 | 54,929 |
| | Finngen | 32 | ICD-10, drug prescription record | 2886 | 215,272 | 218,158 | 11,391 |
| | UKBB | 21 | ICD-10 | 3050 | 196,302 | 199,352 | 12,013 |
| | PHB | 21 | ICD-10, drug prescription record | 1943 | 5723 | 7666 | 5802 |
| | EGCUT | 21 | ICD-10, drug prescription record | 1182 | 15,605 | 16,787 | 4395 |
| | MVP | 22 | ICD-10 | 110,823 | 271,686 | 382,509 | 314,859 |
| | Total | | | 136,867 | 776,327 | 913,194 | 465,415 |
| AFR | AoU | | ICD-10, drug prescription record | 4215 | 26,233 | 30,448 | 14,526 |
| | MVP | 22 | ICD-10 | 47,384 | 47,483 | 94,867 | 94,867 |
| | Total | | | 51,599 | 73,716 | 125,315 | 121,412 |
| Cross-ancestry | Total | | | 188,466 | 850,043 | 1,038,509 | 617,055 |

*AoU* All of Us, *UKBB* UK Biobank, *PHB* Partners HealthCare Biobank, *EGCUT* Estonian Genome Center of the University of Tartu, *MVP* Million Veteran Program.
[a]Summary statistics of a previously published GWAS.
[b]Effective sample size = $4/(1/n_{cases} + 1/n_{controls})$.

## Genetic correlations

We used LDSC to calculate the genetic correlations between EHR-ED (the EUR meta-analysis) and 50 traits, including psychiatric disorders, substance use, behavioral phenotypes, personality traits, and general health characteristics (Supplementary Data 5). EHR-ED was positively significantly correlated with the number of children ($r_g = 0.301 \pm 0.034$), depression ($r_g = 0.285 \pm 0.027$), attention-deficit/hyperactivity disorder (ADHD) ($r_g = 0.278 \pm 0.035$), PTSD ($r_g = 0.277 \pm 0.003$), heart failure ($r_g = 0.259 \pm 0.042$), T2D ($r_g = 0.268 \pm 0.031$), and cannabis use disorder (CanUD; $r_g = 0.25 \pm 0.032$). It was negatively correlated with age of first sexual relations ($r_g = -0.3 \pm 0.026$), and age of smoking initiation ($r_g = -0.248 \pm 0.033$) (Fig. 2, Supplementary Data 5). There were no significant genetic correlations between EHR-ED and any of the 3935 brain measures (Supplementary Data 6).

## Local genetic correlations (LAVA)

We used LAVA to calculate the local genetic correlations between EHR-ED (the EUR meta-analysis) and the 50 traits that were used to calculate LDSC (see above; see Supplementary Data 5). There were 76 local genetic correlations considering all pairs, of which 37 associated EHR-ED with substance use traits. Most prominently, cannabis lifetime use (CanLU) had 12 genetically correlated regions with ED, problematic alcohol use (PAU) had seven, and smoking cessation, smoking initiation, and age of smoking initiation had five, four, and five associated regions with EHR-ED, respectively. In addition, neuroticism was genetically correlated with EHR-ED in eight regions, prostate cancer in seven, and pulmonary hypertension in four. Out of 76 genetic correlations, five regions appeared more than once: the region between chr6:32454578–32539567 correlated CanUD and atrial fibrillation (AF) with EHR-ED; the region between chr6:99678877–100878645, that includes the strongest hit in our GWAS, rs78677597, genetically correlated ED, in a positive direction, with both BMI and obesity; the region between chr8:2070903–2396122, associated CanLU and number of cigarettes per day with EHR-ED; the region between chr11:68887345–69713996 associated prostate cancer and Alzheimer's disease with EHR-ED; the region between chr16:53393883–54866095, associated heart failure and obesity with EHR-ED (Supplementary Data 7).

## Mendelian randomization (MR)

All traits with significant genetic correlations ($r_g$) with EHR-ED in EUR (a total of 30 traits) were analyzed using MR, using a p-value threshold of $1 \times 10^{-5}$ to define genetic instruments (Fig. 3). MR analyses inferred significant causal effects of EHR-ED (as exposure) on 15 traits (and a nominal effect, which did not survive after Bonferroni correction, for 10 more traits; Supplementary Data 8) and a significant causal effect of 22 traits on EHR-ED (as outcome; and a nominal effect for five more traits; Supplementary Data 9). Fourteen traits showed bidirectional causal relationships with EHR-ED. The strongest causal effects of EHR-ED were on T2D ($effect_{corrected} = 0.222 \pm 0.058$, $p_{corrected} = 1.22 \times 10^{-4}$) and obesity ($effect_{observed} = 0.196 \pm 0.032$, $p_{observed} = 8.94 \times 10^{-10}$), and the strongest negative effect was on age of first sexual relations ($effect_{corrected} = -0.241 \pm 0.043$, $p_{corrected} = 2.46 \times 10^{-8}$). The strongest causal effects on EHR-ED were of OUD ($effect_{corrected} = 0.297 \pm 0.088$, $p_{corrected} = 7.28 \times 10^{-4}$) and CanUD ($effect_{corrected} = 0.228 \pm 0.038$, $p_{corrected} = 3.15 \times 10^{-9}$).

## PheWAS

rs78677597 was more significantly associated with EHR-ED in EUR than any other variant by at least 120 orders of magnitude ($p = 5.35E-139$ vs. *PHF21B**rs8141413, the second strongest hit, with $p = 4.18E-19$), and therefore this was the focus of our PheWAS. After Bonferroni correction, 15 traits out of 330 were significantly associated with rs78677597, nine of which are directly related to cardiovascular and metabolic conditions (Fig. 4, Supplementary Data 10). The other traits were pain, dermatological infections, and respiratory conditions. For 14 traits, rs78677597 had a positive effect, whereas the only trait with which it was negatively associated was disorders of lipoprotein metabolism and lipidemias. Five traits—atrial fibrillation and flutter, obesity, cardiac arrhythmias, heart failure, and joint pain—were associated with rs78677597 with a lower *p*-value than the standard threshold for significance in GWAS ($p < 5 \times 10^{-8}$).

## TWAS

We used TWAS to evaluate predicted changes in differential gene expression in EUR. We identified nine independent associated genes in nine different tissues (Supplementary Table 6). The most significant gene was *CTNNB1* ($p = 6.8 \times 10^{-9}$) with negative enrichment in the amygdala. Of the nine genes, one (*TEX41*) was also GWS.

## SMR

Two genes had a significant effect in the entire brain: *CTNNB1* had a negative effect ($\beta = -0.091$, $p = 1.79 \times 10^{-7}$), suggesting that the genetic variants associated with brain expression this gene are negatively associated with EHR-ED; *C1GALT1* had a positive effect ($\beta = 0.051$, $p = 3.63 \times 10^{-7}$), suggesting that the genetic variants associated with brain expression of this gene are positively associated with EHR-ED (Supplementary Data 11).

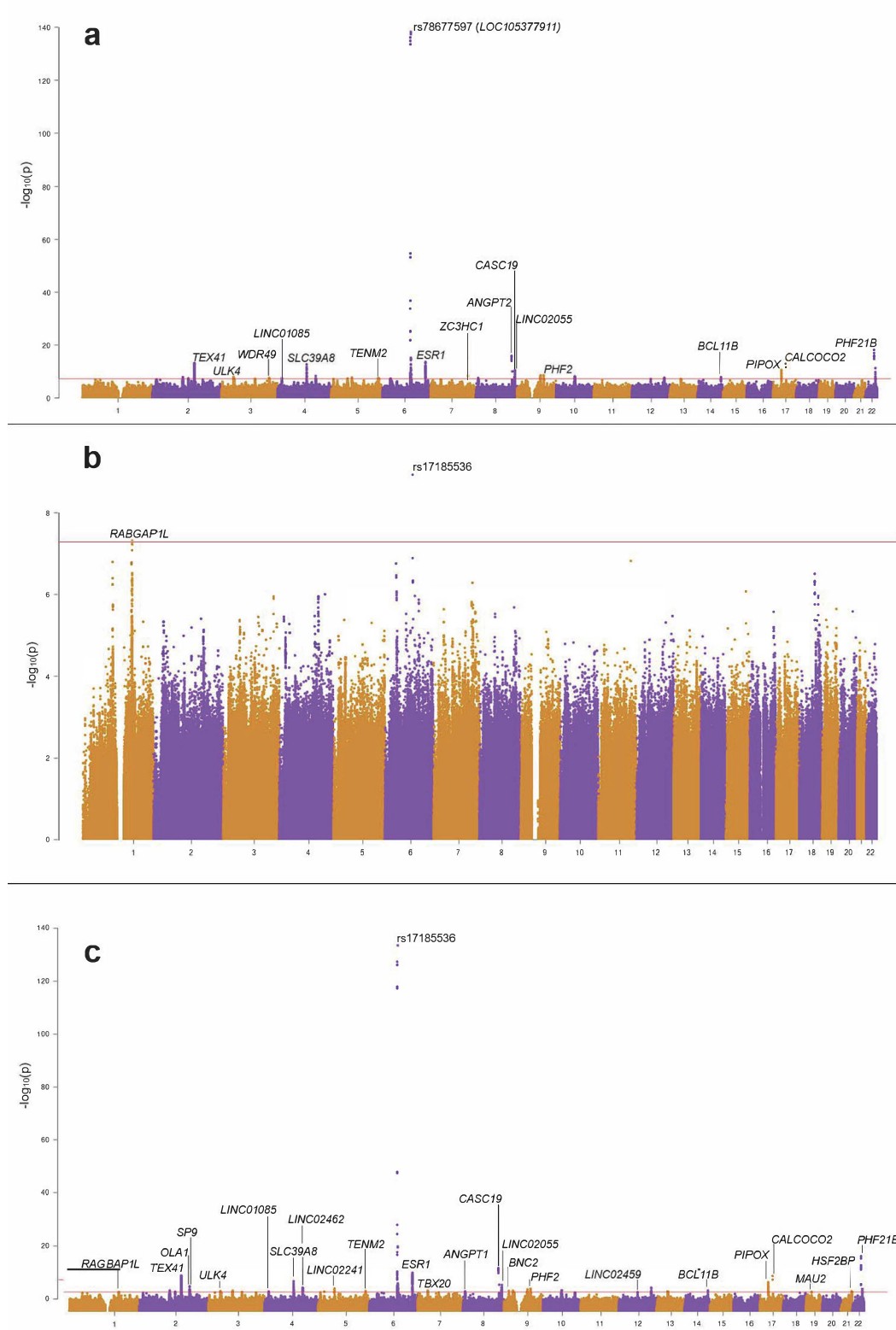

**Fig. 1 | EHR-ED GWAS.** GWAS meta-analysis of EHR-defined erectile dysfunction (EHR-ED) in (**a**) EUR population ($n_{total}$ = 913,194, $n_{eff}$ = 465,415), (**b**) AFR population ($n_{total}$ = 125,315, $n_{eff}$ = 121,412), and (**c**) cross-ancestry (EUR-AFR; $n_{total}$ = 1,038,509, $n_{eff}$ = 617,054). Variants that are located within or near genes are annotated. Statistical significance is defined by the standard $p$-value threshold of $5 \times 10^{-8}$.

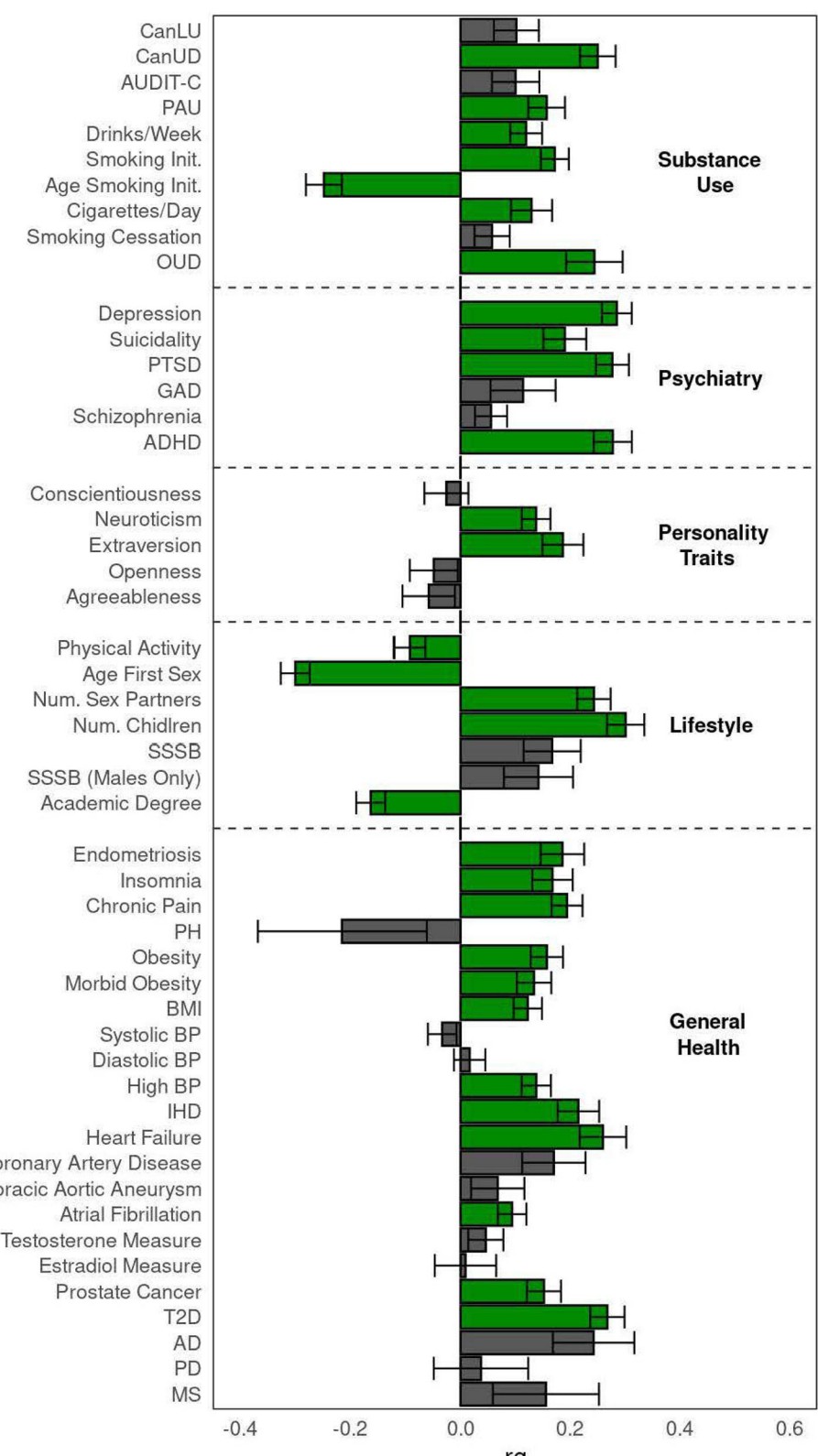

**Fig. 2 | Genetic correlations between EHR-ED and 50 traits of interest.** Traits with statistically significant correlations are highlighted in green. Significance threshold is corrected for 50 tests using Bonferroni correction ($p < -0.001$). CanLU cannabis lifetime use, CanUD cannabis use disorder, AUDIT-C alcohol use disorders identification test -consumption, PAU problematic alcohol use, OUD opioid use disorder, PTSD post-traumatic stress disorder, GAD generalized anxiety disorder, ADHD attention-deficit/hyperactivity disorder, SSSB same-sex sexual behavior, PH pulmonary hypertension, BMI body mass index, BP blood pressure, IHD ischemic heart disease, T2D type 2 diabetes, AD Alzheimer's disease, PD Parkinson's disease, MS multiple sclerosis.

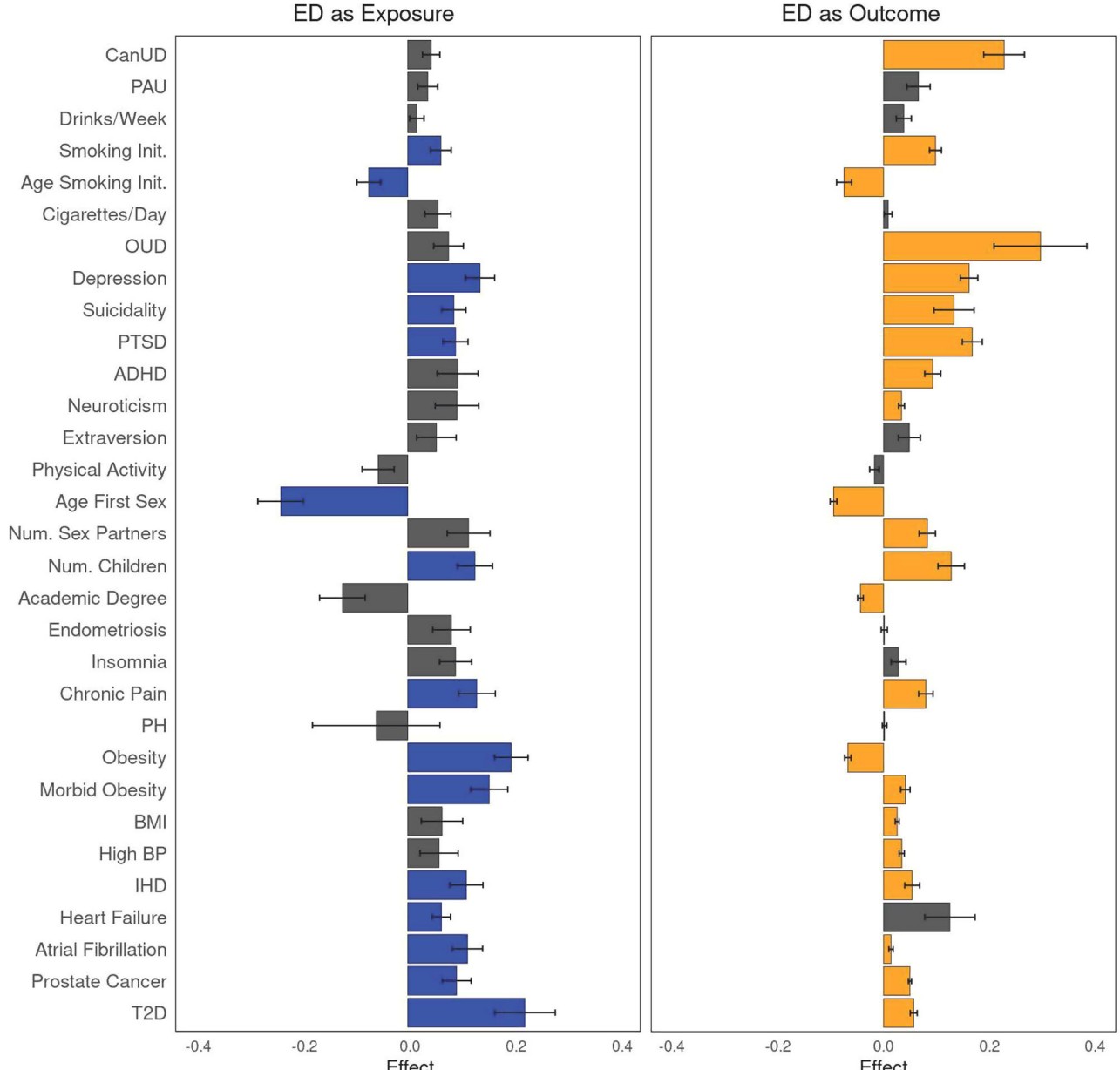

**Fig. 3 | Causal genetic relationship between EHR-ED and various traits, with EHR-ED as exposure (left, blue) and outcome (right, orange).** Gray bars indicate non-significant effects after Bonferroni correction ($p < 0.0017$). CanUD cannabis use disorder, PAU problematic alcohol use, OUD opioid use disorder, PTSD post-traumatic stress disorder, ADHD attention-deficit/hyperactivity disorder, BMI body mass index, BP blood pressure, IHD ischemic heart disease, T2D type 2 diabetes.

## Drug repurposing

For drug repurposing analysis, we created a list of 67 genes that had a significant effect in the GWAS and/or MAGMA and/or TWAS analyses in EUR, AFR, and cross-ancestry (Supplementary Data 12). We found 14 drugs that have a potential drug–gene interaction in our study. Twelve of those were associated with *ESR1* (which encodes estrogen receptor 1) while the other two were associated with *CTNNB1* and *ANGPT2* (Supplementary Table 7).

## gSEM

gSEM was performed to examine the overarching genetic relationships between EHR-ED and 12 traits of interest that had demonstrated significant genetic correlations with EHR-ED. EFA suggested a three-factor model fit the data best, explaining 57.2% of cumulative variance (Supplementary Table 8). Factor 1 (SS loading: 3.16) explained 24.3% of the variance, Factor 2 (SS loading: 2.29) explained 17.6%, and Factor 3

(SS loading: 1.98) explained 15.2% of the overall variance. Traits with loadings > 0.25 were evaluated on the respective factors in CFA. CFA suggested the three-factor model fit the data well via traditional fit indices, including a comparative fit index of 0.93 and a standardized root mean square residual of 0.07 (Supplementary Data 13). EHR-ED co-loaded on Factor 1 with SUDs and related traits (OUD, CanUD, PAU, number of lifetime sex partners, ADHD) and Factor 3 with physical health traits (T2D, high blood pressure, ischemic heart disease, and obesity). PTSD, depression, and endometriosis loaded solely on Factor 2, along with ADHD, which also co-loaded on Factor 1. Factor 2 demonstrated moderate correlations with both Factor 1 (0.57) and Factor 3 (0.48). Factors 1 and 3 were correlated at 0.12 (Fig. 5).

## Discussion

ED is an important clinical problem that affects quality-of-life and reproductive capacity. Little is known about the genetic factors that

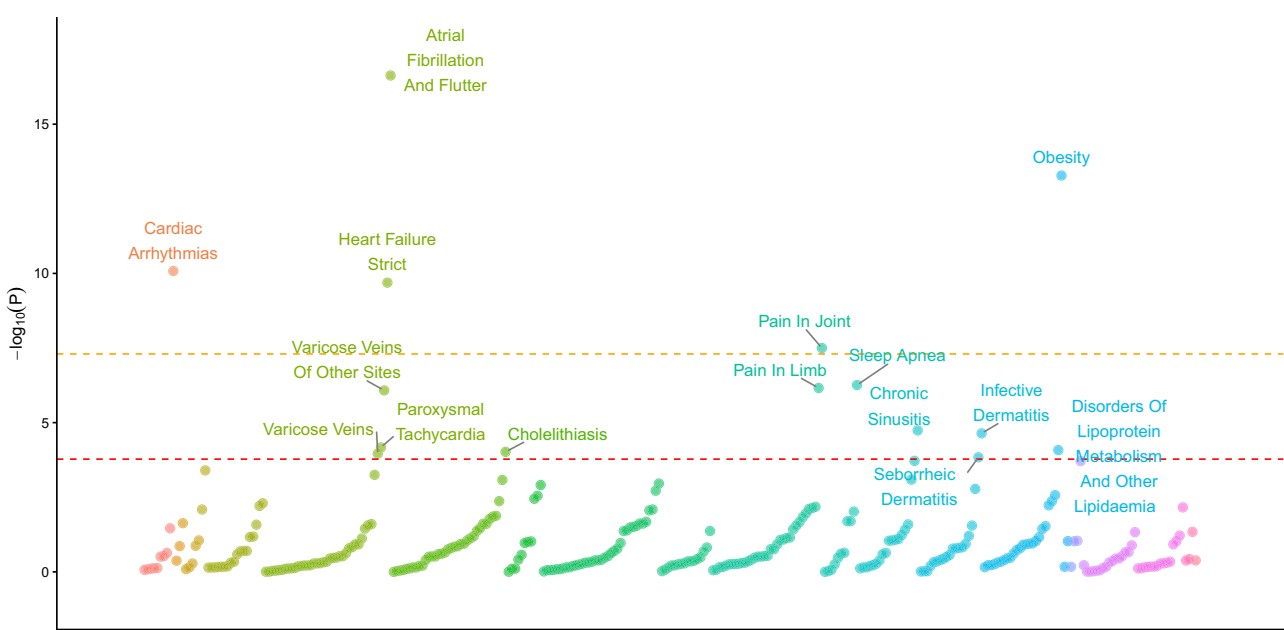

**Fig. 4 | pheWAS (two-sided linear/logistic regression across traits) of rs78677597 among 330 traits from the Finngen, UKBB, and MVP meta-analysis.** The red line represents the significance threshold after Bonferroni correction ($p = 1.67 \times 10^{-4}$). The orange line represents the standard GWAS significance threshold ($p = 5 \times 10^{-8}$). Traits with a significant effect are annotated. The direction of each triangle indicates the effect direction.

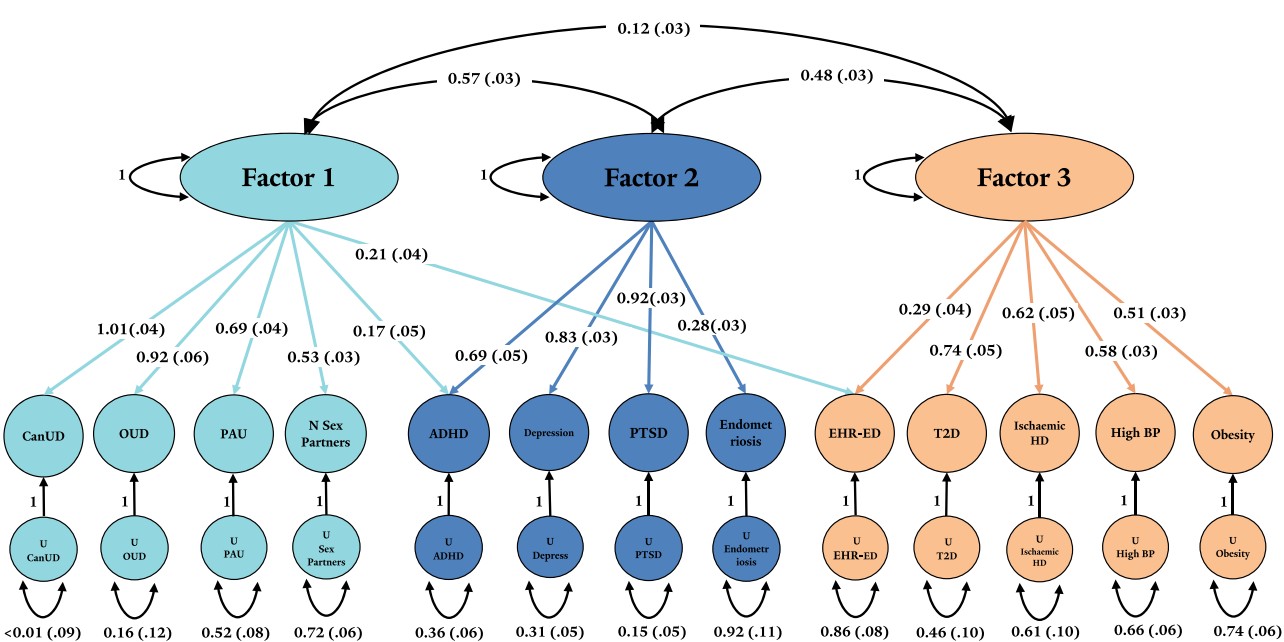

**Fig. 5 | Genomic structural equation modeling (gSEM) assessing the overarching genetic relationship between EHR-ED and 12 traits of interest.** CanUD cannabis use disorder, OUD opioid use disorder, PAU problematic alcohol use, ADHD attention-deficit/hyperactivity disorder, PTSD post-traumatic stress disorder, EHR-ED electronic health record erectile dysfunction, T2D type 2 diabetes, IHD ischemic heart disease, BP blood pressure.

affect risk for this disorder. In our analysis of the AoU sample, we found one GWS locus in EUR, rs17185536, which is located in a non-coding region on chromosome 6. This result replicates several previous ED GWAS[20–22]. In AFR, in AoU, we did not find any significant SNPs. In meta-analyses of EUR populations, AFR populations, and a cross-ancestry, we found numerous variants associated with EHR-ED. In total, we found 40 lead SNPs in EUR, mapping close to or within 19 coding and non-coding genes, and two lead SNPs in AFR, of which one is intronic to a coding gene. In the cross-ancestral analysis, we found 51 lead SNPs, mapping close to or within a total of 26 different coding and non-

coding genes. PRS analysis in EUR revealed that our GWAS explains 9.2% of the variance in EHR-ED.

This study improved on prior knowledge of associated genes by 50%, adding five novel coding genes associated with EHR-ED in EUR compared to the best-powered previous study[22], which revealed a total of nine. In AFR, there was no increment to the two significant SNPs found previously[22], though there was a novel significant variant, mapping to *RABGAP1L* on chromosome 1 (with one significant SNP lost in the meta-analysis). Cross-ancestry, we found a total of 26 genes associated with EHR-ED, the majority of them novel. In addition, this

study provides a scope of novel post-GWAS analyses and adds important new insights regarding the genetic architecture of ED and its association with a variety of physiological and psychiatric traits, including possible risk factors for ED, such as cannabis and opioid use disorders.

The most robust effects were of rs78677597 in EUR ($p = 5.35 \times 10^{-139}$), and of rs17185536 in AFR ($p = 1.17 \times 10^{-9}$) and cross-ancestry ($p = 5.30 \times 10^{-138}$), replicating previous findings of a strong association between this region on chromosome 6 and ED[20–22] (rs78677597 and rs17185536 are located in the same LD region). The rs78677597 variant is located within the non-coding region *LOC105377911*, and has been implicated as an inhibitor of *SIM1*, a gene located ~230 kb apart from this variant on chromosome 6. Data obtained from the GTEx portal ([gtexportal.org](gtexportal.org)) indicated that rs17185536, mapping 11 kb from rs78677597 and with very high LD ($r^2 = 0.97$), is an eQTL for *SIM1*, with a negative effect on its expression. rs17185536 was previously identified as the lead SNP in a study of ED[20] and is potentially the functional variant in this region, affecting *SIM1* expression[20,33]. The fact that different lead SNPs were identified across ancestries likely reflects differences in LD patterns. The study that identified rs17185536 as the lead SNP was cross-ancestral[20], which aligns with the fact that the same SNP was the lead variant in our cross-ancestry meta-analysis, too. Literature regarding *SIM1* mostly associates it with obesity: animal studies show that downregulation of SIM1 is associated with hyperphagia and weight gain[24,34]; in humans, rare genetic mutations in and around *SIM1* were found to be associated with childhood obesity[35,36]. *SIM1* is also implicated in the effect of melano-cortinergic pathways on sexual function and behavior (melanocortin is an important mediator of sexual function, and specifically penile erection[37]): impaired sexual behavior in *mc4r* (melanocortin receptor 4)-knockout male and female mice was normalized by expressing mc4r exclusively on sim1 neurons[38,39]. In the pheWAS analysis, we found that rs78677597 is significantly associated with obesity, as well as with eight other traits related to cardiovascular and metabolic conditions. The other traits that were found to be affected by rs78677597 in the phe-WAS—joint pain, limb pain, sleep apnea, chronic sinusitis, infective dermatitis, and seborrheic dermatitis—are themselves all associated with weight gain and obesity[40–45]. Cardiovascular traits, obesity, and EHR-ED also loaded under the same factor in the gSEM analysis. Local genetic correlation analysis in EUR showed that both obesity and BMI (but not other traits) were correlated with EHR-ED through a region on chromosome 6 that includes rs78677597 and *SIM1*. Indeed, there is high comorbidity between ED and obesity (79%[7]). Our findings suggest that these two traits share a common genetic basis, perhaps in large part through the region tagged by rs78677597 and its effect on *SIM1* expression. In the MR analysis, we found that having genetic variants which are associated with EHR-ED is a much stronger causal factor for morbid obesity than having genetic variants associated with morbid obesity is causal for EHR-ED; obesity (not necessarily morbid) was found to be caused by genetic liability to EHR-ED but negatively correlated to, i.e., associated with reduction of, EHR-ED. These findings indicate the important role of variants associated with EHR-ED in obesity and morbid obesity. These results do not mean that ED causes obesity per se (i.e., MR tells us about genetic, not phenotypic, causality). These results are consistent with the interpretation that rs78677597 has a stronger effect on EHR-ED than on obesity or any other studied trait, suggesting that EHR-ED is not merely a symptom of obesity, but a unique trait that is associated with obesity, i.e., with common causality.

Variants that do not map to the chromosome 6 region are all at least 100 orders of magnitude less strongly associated with EHR-ED than the lead region, and accordingly, their contributions to EHR-ED risk are very much lower, taken individually. Several genes that had a significant effect on EHR-ED in our study are mostly known for their associations—according to previous GWAS publications—with

cardiovascular traits: *ULK4* was associated with pulse pressure measurement and diastolic blood pressure (BP)[22,46]; *WDR49* with platelet volume and platelet count[47,48]; *ZC3HC1* with platelet count, as well as with coronary artery diseases[22,49]; *ANGPT2* with diastolic BP[50–52]; *TBX20* with heart development and function[53,54]; and *HSF2BP* with both systolic and diastolic BP[46,55]. *TEX41* (testis-expressed 41) was strongly associated with a variety of traits, such as diastolic BP[46] and also smoking initiation[56]. In the TWAS, *TEX41* was found to be in low enrichment in the testis, providing further indication of its possible role in sexual functioning. This finding finds context in literature regarding the role played by TEX genes in male fertility, but not in sexual functioning[57]. Normal-range testicular function is generally not associated with ED.

The non-coding RNA *CASC19* also had a strong effect on EHR-ED. Several studies associate this gene, and specifically *CASC19*\*rs138042437 (lead SNP for this locus in our study), with prostate cancer[58–61], a trait we found to have a bidirectional causal relationship with EHR-ED, in accordance with known strong comorbidity between the two traits[62,63] and with increased prevalence of prostate cancer among ED subjects in our sample as well. This association is likely causal, as both prostate cancer and prostate cancer treatments may lead to ED in patients[62,63]. Even though the genetic correlation between prostate cancer and EHR-ED was relatively low ($r_g = 0.152 \pm 0.031$), there were eight local genetic correlations between these traits, all in a positive direction. The strongest of those was for the region on chromosome 8 that includes *CASC19*. Another significant genetic correlation between these traits can be seen on chromosome 17, in a region that includes *CALCOCO2*, which was GWS for EHR-ED in our main analysis, and was also previously associated with prostate cancer[32].

One of the highest non- rs78677597-associated GWS regions in our study was of *ESR1*\*rs2347923 in EUR ($p = 2.42 \times 10^{-14}$) and *ESR1*\*rs6557171 cross-ancestry ($p = 2.79 \times 10^{-15}$), coding for estrogen receptor α (ER-α), one of the two main estrogen receptors. Although they are mainly known as female sex hormones, estrogens play vital roles in various processes in males too[64]. Overexpression of estrogen and 17β-estradiol is associated with ED, both in rodents[65,66] and humans[67]. A study in rats showed that overexpression of ER-α does not impair fertility, but significantly lowers the weight of bulbospongiosus and levator ani muscles[68], which play an important role in erection and sexual functioning[69]. It is possible that changes induced by variations in this gene lead to over-expression of *ESR1* or to stronger affinity between ER-α and its ligand 17β-estradiol, which both can lead to ED. The role played by ER-α in prostate cancer[70], alongside the high comorbidity seen between prostate cancer and ED[71], raised the concern that the effect seen in *ESR1* is driven by prostate cancer and not directly by EHR-ED; however, previous GWAS publications indicate that the effect of *ESR1* on prostate cancer is low[72] or null[22,32]. Most of the drugs we identified in the drug repurposing analysis are ER-α-antagonists, which makes them potential candidates for ED treatment. Toremifene can act as an agonist or antagonist of estrogen[73], and it successfully lowered prostate cancer rates in men with intraepithelial neoplasia, probably due to its antiestrogenic properties in the prostate, with no effect on ED[74,75]. In a study in healthy elderly men, raloxifene increased luteinizing hormone (LH), follicle-stimulating hormone (FSH), and sex steroid hormones, with no negative effects on erectile function[76]. The gene encoding the FSH receptor was previously associated with ED following radiotherapy for prostate cancer in African Americans[77]. On the other hand, in people who suffer from ED, LH levels were significantly higher[78], suggesting that raloxifene may not be a suitable medication for ED. The same study showed no association between testosterone levels and ED[78], in line with the non-significant $r_g$ we found between the two traits. Tamoxifen, also an estrogen antagonist, was found to elevate ED in male breast cancer patients[79]. An explanation for that may arise from tamoxifen-induced

enhancement of sex hormone binding globulin, which limits the amount of free estradiol, but can also lead to ED[80]. Other ER-α-antagonists that were identified for potential drug repurposing are fulvestrant and danazol, but we found no data regarding their effects on men's sexual functioning.

*ESR1* is also associated with anxiety and depression[81,82], which we found to be genetically correlated with EHR-ED ($r_g = 0.285 \pm 0.027$). *ESR1*\*rs2347923 and *ESR1*\*rs6557171 had significant effects on depression, mood disorders, and the use of antidepressants in Finngen participants[32]. A recent GWAS in AoU and UKBB subjects showed that the effect of *ESR1* on depression may be specific to males[83]. Endometriosis, on the other hand, is a female-specific trait that is also affected by *ESR1*[32] and is associated with sexual dysfunction[84]. In consideration of its somewhat congruent relationship to sexual dysfunction in men, we sought to discover whether this trait is genetically correlated with EHR-ED, and found a significant genetic correlation between the traits, suggesting shared genetic architecture between these sex-specific traits. Nevertheless, these traits did not load under the same factor in the gSEM analysis, in which endometriosis loaded alongside traits of a psychiatric nature, such as depression and PTSD.

*SLC39A8*\*rs13135092, which encodes the ZIP8 zinc transporter, had a significant association with EHR-ED as well. Along with the effects found in the gene-based analyses for several genes that are associated with zinc finger formation (*ZKSCAN3*, *ZKSCAN4*, *ZKSCAN8*, *ZSCAN16*, and *ZSCAN31*), this outcome corroborates findings regarding an important role of zinc in erectile function[85,86]. In addition, *SLC39A8* was previously found to be upregulated in rats subjected to a diabetic ED model[87], consistent with a known genetic and phenotypic association between ED and T2D[3,21], which was also seen in our data (see Supplementary Table 1). Our study, too, supports a bidirectional genetic correlation between EHR-ED and T2D, with T2D being the trait most strongly positively and causally affected by EHR-ED out of all the traits we tested. In humans, *SLC39A8* was associated with testicular dysfunction[22], and had also a strong association with high-density lipoprotein (HDL) cholesterol levels[22,88], obesity[22], and various brain morphology and brain volume phenotypes[89–91]. Specifically, it was significantly associated with the volume of the amygdala[92,93], with rs13107325 (-9 kb apart from rs13135092) having the strongest−negative−effect on the right-central amygdala volume ($p = 4.47 \times 10^{-101}$)[93]. Furthermore, *CTNNB1*, which encodes β-catenin, was one of the most strongly associated genes with EHR-ED in the gene-based analysis; it was associated with low enrichment in the amygdala (in the TWAS) and in the entire brain (in the SMR) of EHR-ED subjects, suggesting that β-catenin downregulation in the brain may increase ED risk. *CTNNB1* is adjacent to and partially overlaps with *ULK4*, which was GWS both in EUR and cross-ancestry, and these genes likely represent the same GWS locus in our results. In animal studies, β-catenin was suggested as an important factor in the development of erectile tissues[94]. Specifically in the amygdala, lower β-catenin was associated with anxiety and depressive-like behavior[95]. In total, these findings suggest that *SLC39A8* and *CTNNB1* might contribute to psychogenic ED through changes invoked in the amygdala, and provide a possible mechanism for the moderate genetic correlations we found for EHR-ED with depression and PTSD. We did not, however, find significant genetic correlations between EHR-ED and a variety of brain volume and connectivity measures.

In the drug repurposing analysis, the nonsteroidal anti-inflammatory drug (NSAID) sulindac was identified as a possible treatment for ED through its targeting of β-catenin. Most studies suggest that sulindac is a suppressor of β-catenin expression, a trait of this drug that was investigated mostly in cancer (for example, refs. 96,97). However, sulindac is also an inhibitor of PDE5, mimicking the action of common anti-ED drugs by elevating cGMP (this study was conducted in cell cultures)[98]. Sulindac, therefore, may provide an interesting focus for potential drug repurposing, considering the

evidence of its potential to upregulate brain β-catenin and to inhibit PDE5 activity.

*PHF21B* was GWS (rs8141413, $p = 4.18 \times 10^{-19}$ in EUR, $p = 2.63 \times 10^{-21}$ cross-ancestry), and had also the strongest effect of all genes in the gene-based analysis in EUR ($p = 1.24 \times 10^{-10}$). Besides ED[22], *PHF21B* is associated with MDD[99,100] and is over-expressed in prostate cancer[101]. Moreover, it was found to be associated with the lifetime number of opposite-sex sexual partners[102]. Counterintuitively, EHR-ED in our study had a positive genetic correlation with the number of sexual partners and number of children, and a negative correlation with the age of first sexual intercourse. Even more surprising were the bidirectional causal relationships between these traits, especially with a lower age of first sexual relations, a relatively strong outcome of the genetic liability to suffer from EHR-ED. The MR results showing that EHR-ED is causal for the age of first sexual relations is obviously not a phenotypic causality, but a genetic one, i.e., genetic inclination to develop ED in the context of seeking treatment for it, even if later in life, is causal for genetic inclination to lower age of sexual relations. One possible explanation is a positive correlation between sexual drive and the available ED phenotype, with high sexual desire among people who complain of ED[103,104]. Another study found that ED alone was not enough to drive men to seek treatment, but that it had to be in conjunction with sexual desire[104]. ED is usually a self-reported trait; even though the phenotype definition in our study was based on EHR, the indication of ED was most likely made based on patients' self-reports to their physician. These are likely to be subjects who are or want to be sexually active, hence the strong genetic correlation between ED and active sexual life, which is seen in lower age of sexual relations, number of lifetime sexual partners, and arguably even number of children. The main cause for under-reporting of ED is embarrassment[31], and a 2007 study found that 19% of Taiwanese participants over 40 who did not self-report ED suffered from ED symptoms[30]. A positive $r_g$ between EHR-ED and extraversion is another indicator of the less shy nature of the people who report to ED. This may reflect the difference between EHR-ER and ED per se, i.e., it is an indication of possible ascertainment bias.

We found significant genetic correlations between EHR-ED and several substance use traits, most prominently CanUD ($r_g = 0.25 \pm 0.033$), OUD ($r_g = 0.244 \pm 0.051$), and age of smoking initiation ($r_g = -0.248 \pm 0.033$). Several genes that were identified in our study are associated with smoking initiation, amongst them are *RABGAP1L*, *OLA1*, *PIPOX*, *BCL11B*, and *TEX41* (the first two are also associated with the number of alcoholic drinks per week)[56]. EHR-ED had a genetic correlation with CanUD and PAU, but there were no significant correlations with CanLU and alcohol use disorders identification test−consumption (AUDIT-C) (though the former had 12 local genetic correlations with EHR-ED and the latter had two); AUDIT-C and CanLU are, respectively, traits of alcohol and cannabis use, while CanUD and PAU are traits closer to dependence on the same substances[105–107]. Moreover, the causal relationship of EHR-ED with CanUD and OUD was unidirectional, with both of these traits among the strongest causal factors for EHR-ED (of the traits included in the study), indicating that genetic risk for cannabis and opioid dependence, but not necessarily non-dependent use, are relatively strong risk factors for developing ED. These results are in line with well-established associations between ED and cannabis and opioid use and abuse[17,108,109]. Furthermore, gSEM analysis revealed that different risk-taking behaviors, such as substance use disorders and lifetime number of sex partners, loaded under the same factor along with EHR-ED.

This study is the best-powered so far to examine the genetic architecture of ED. The strongest effects were found for the non-coding rs78677597 in EUR and rs17185536 in AFR, and cross-ancestry confirm, with much greater power, previous findings regarding the unique role of this region and its strong effect on ED[20–22], suggesting a prominent biological role for *SIM1*. Genes identified in this study

mostly point to cardiovascular factors, prostate- and testis-related genes, and several genes that affect both hormonal homeostasis and brain function. The biology revealed in this study is consistent with other indications that ED is not merely a consequence of penile physiological malfunction, but is a complex trait that can also be influenced by substance use, hormonal changes, and psychological state. The fact that ED loaded under two factors in the gSEM analysis—one associated with risk-taking and substance use, and the other with cardiovascular and metabolic traits—reflects its complex etiology. Genetic correlations between EHR-ED and several psychiatric, cardiovascular, personality, lifestyle, and general health traits provide further indication of this complexity. Our findings provide novel targets for possible treatment for ED, and emphasize the need to investigate *SIM1* deeply to understand the mechanism through which it affects erection. Considering that common anti-ED drugs are not effective for 30–35% of the patients[29], these new targets are of great need and importance.

This study has limitations. First, as discussed, ED is mainly a self-reported trait. Some subjects experienced ED but never reported it, and were therefore assigned as controls instead of cases. The trait we studied—EHR-ED—includes subjects with ED who were, generally, distressed enough about it to seek medical attention, although some cases may have been brought to light by routine history taking. Those who receive an ED diagnosis are likely to be patients who want treatment for this condition, i.e., who wish to have sexual relations. This would be expected to decrease power for these analyses rather than lead to false positives in gene identification, but the psychological concomitants of reporting ED or failing to report it may have influenced the genetic correlations we detected. Second, we defined ED on a case-control basis, even though there are different levels of severity of ED. If severity data were available in the future on a large scale, it would allow for more accurate and possibly more powerful analysis of the ED. Treatment-resistant ED would be of particular interest. Third, we aimed to conduct our analysis in AoU as similarly as possible to the already-completed analyses in other cohorts included in the meta-analysis. For this reason, we did not include or exclude subjects according to the number of inpatient/outpatient visits (i.e., one visit was enough to include a subject as an ED case). Similarly, we did include an age limit for the analysis, although ED is more common in older ages (we did, however, use age as a covariate in the regression), nor did we exclude subjects with a history of prostate cancer treatment, which is a risk factor for ED[71]. Considering the relatively large proportion of prostate cancer history among ED subjects in our analysis (17.8%, compared to 5.3% in the control group in EUR; 16%, compared to 1.9% in the controls group in AFR), it is plausible that inclusion of prostate cancer subjects could affect the results; for example, some of the genes we identified in this study, *ESR1* and *CASC19*, are genome-wide significantly associated with prostate cancer too[58–61,70,72,101], and therefore our results may be driven in part by these associations. These inclusion criteria enabled us to meta-analyze phenotypically similar traits, but at the potential cost of increased noise. Nevertheless, in terms of the strongest GWS locus, our main analysis achieved similar results to previous ED GWAS (with increased power), which either included[21,22] or excluded[20] subjects with a history of prostate cancer, suggesting that our results are valid beyond intervening variables. Fourth, case/control distribution in our samples varies widely, which may be associated with age distribution: in MVP, a relatively older sample (mean age 62)[22], the prevalence of ED was 28.9%, compared to 19.1% in the younger cohort of AoU (mean age 58). Nevertheless, in Finngen, only 1.32% of the subjects were ED cases, even though the mean age in this cohort (61) is closer to that of MVP[32], suggesting that other factors, such as diagnosis method, are of importance as well. These differences would have increased between-sample heterogeneity. Fifth, genetic correlations were mostly calculated vs traits that were measured in samples of both males and females, because male-only data are often not available, and if available, are less powerful. This does not apply to the genetic correlation analysis with endometriosis, which is a female-only trait. Therefore, this is an analysis of a more exploratory nature, and the genetic correlations found between these traits cannot be directly asserted phenotypically. Sixth, a low sample size in the AFR cohort was likely a main reason for the low number of significant results in this ancestry. Future studies using larger sample sizes are therefore still in need.

## Methods

This research was not restricted or prohibited in the setting of any of the included researchers. AoU was approved by an AoU ethics review committee. We do not believe our results will result in stigmatization, incrimination, discrimination, or personal risk to participants. No statistical method was used to predetermine sample size. No data were excluded from the analyses.

### Genome-Wide Association Study (GWAS) analysis and meta-analysis

For our primary analysis, we used data from the All of Us (AoU) biobank, version 8 (GRCh38-built). Genotyping, quality control measures, and ancestry assignment were described previously[110]. EHR-ED was defined according to (a) participants' electronic health record (ICD-10 code N52) or (b) participants being prescribed one or more of the ED drugs (PDE5-inhibitors), sildenafil (when prescribed as Viagra; this same medication is also used to treat pulmonary hypertension under a different brand name), tadalafil, and vardenafil. Subjects who met at least one of these criteria were included in the analysis as cases. PDE5 inhibitors are also prescribed to treat pulmonary hypertension; therefore, participants who suffer from pulmonary hypertension according to their EHR (~5% of the study population) were excluded. Females were excluded from the analysis because the ED trait is male-specific. We conducted separate analyses for subjects of genetically defined EUR and AFR ancestries[110]. After removing one individual from each pair of related subjects (kinship coefficient cutoff = 0.1)[110], we reached a final sample size of 88,722 EUR (16,983 cases) and 30,448 AFR (4215 cases) subjects. GWAS was conducted using logistic regression in PLINK 2.0, with the first ten genetic PCs and age as covariates. Variants with minor allele frequency (MAF) < 0.1% and Hardy–Weinberg equilibrium (HWE) $p < 1 \times 10^{-6}$ were excluded. We then conducted three separate meta-analyses: within-EUR, within-AFR, and cross-ancestry, including our results from AoU combined with previously published data, gained through the following cohorts: UK Biobank (UKBB), the Estonian Genome Center of the University of Tartu cohorts, hospital-recruited Partners HealthCare Biobank (PHB) cohort (6175 EUR cases)[21], the Million Veteran Program (MVP) (110,823 EUR and 47,384 AFR cases)[22], and Finngen (2886 EUR cases)[32]. Cases in these cohorts were defined by ICD-10 code, drug prescription records, and self-report. Within-EUR, within-AFR, and cross-ancestry meta-analyses were performed using METAL[111], weighted by effective sample size (Table 1). In all analyses, we applied a standard genome-wide multiple testing correction ($p < 5 \times 10^{-8}$) to identify genome-wide significant (GWS) loci. The association results were visualized using the R package qqman[112].

### FUMA and MAGMA gene-based and gene set analyses

We used the FUMA platform[113] to identify lead SNPs and genomic loci in our GWAS results. Gene-based analysis was conducted using MAGMA[114], implemented in the FUMA platform. Input SNPs were mapped to 19,028 protein-coding genes in EUR (after Bonferroni correction for 19,028 tests, the statistical significance threshold was set at $p = 2.63 \times 10^{-6}$), 17,515 in AFR (statistical significance threshold: $p = 2.85 \times 10^{-6}$), and 19,118 in cross-ancestry (statistical significance threshold: $p = 2.61 \times 10^{-6}$).

## Heritability estimates and genetic correlations (LDSC)

We performed linkage disequilibrium score (LDSC)[115] regression based on the linkage disequilibrium (LD) reference from the 1000 Genomes phase 3 data, and calculated SNP-based heritability ($h^2$) for the independent cohorts and the meta-analysis. We investigated the inter-cohort genetic correlations, as well as genetic correlations between the meta-analyzed EHR-ED cohort and 50 psychiatric disorders, substance use traits, behavioral phenotypes, personality traits, and general health characteristics, which are associated with cardiovascular function, obesity, hormonal activity, and impaired nitric oxide (NO) bioavailability[22,32,56,58,107,116–134]. After Bonferroni correction for 50 tests (0.05/50), the statistical significance threshold was set at $p = 0.001$. We also calculated the genetic correlations between EHR-ED and 3935 brain structure and function measures[89]. Because many brain measures are not independent from one another, we applied false discovery rate (FDR) correction for multiple tests.

## Local genetic correlations (LAVA)

We used LAVA[135] to calculate local genetic correlations between the EHR-ED EUR meta-analysis and 50 traits of interest, mentioned in the previous section. In both sets of analysis, the genome was divided into 2495 genomic regions to provide minimal LD between the regions and maintain an approximately equal size of the regions of ~1 MB. Breakpoints between regions were computed according to the LD between neighboring SNPs as described previously[135], to maintain regions as relatively independent. Univariate local correlations were calculated for each trait. Bonferroni correction for 2495 regions (0.05/2495) yielded a statistical significance threshold of $p = 2.00 \times 10^{-5}$ for local genetic heritability. Only regions that reached significance were used to calculate genetic correlations among the cohorts (33,442 in 50 pairs. Bonferroni correction for 33,442 tests (0.05/33,442) yielded a statistical significance threshold of $p = 1.49 \times 10^{-6}$.

## Mendelian randomization (MR)

We conducted MR analyses to estimate the causality between EHR-ED and 30 traits that had a significant genetic correlation with EHR-ED when measured using LDSC. All MR analyses were conducted using MRlap[136], which is appropriate for MR analysis with potentially overlapping cohorts. We ran the inverse variance weighted (IVW) model, with a $p$-value threshold of $1 \times 10^{-5}$ to select genetic instruments. Instrument pruning was conducted based on an LD threshold of 0.05, after which the average number of variants for EHR-ED as exposure was 251. For every analysis, MRlap performs a correction for overlapping samples and other potential biases, such as outliers, and presents the statistical difference between the observed and corrected values. Where the difference between these values was significant ($p < 0.05$), we presented the corrected values in the results section; where the difference was not significant, we presented the observed values. Bonferroni correction for 30 tests (0.05/30) yielded a statistical significance threshold of $p = 0.0017$.

## Polygenic risk scores (PRS) analysis

We conducted a leave-one-out meta-analysis of EUR cohorts, excluding AoU subjects (in AFR, there was no need to conduct a separate limited meta-analysis, since the full meta-analyzed data included only two cohorts; see Table 1). Using PLINK 1.9, we calculated the PRS for EHR-ED separately in EUR and AFR, with AoU subjects as the target set in both ancestries. Age and the first 10 genetic PCs were used as covariates. We performed $p$-value-informed clumping with a distance threshold of 250 kb and $r^2 = 0.1$. Risk scores were calculated for a range of $p$-value thresholds ($p \le 1 \times 10^{-6}$, $1 \times 10^{-5}$, $1 \times 10^{-4}$, $1 \times 10^{-3}$, 0.01, 0.05, 0.1, 0.2, 0.3, 0.4, 0.5). Logistic regression was used to test for the association between the PRS scores and ED in the target sets. Bonferroni correction for 22 tests (0.05/22), which accounted for 11 tests in

each ancestry, yielded a statistical significance threshold of $p = 0.0017$. The proportion of variance explained by PRS was estimated using Nagelkerke's pseudo-$r^2$, which is appropriate for binary traits. EHR-ED odds were tested in individuals in the top 2.5%, 5%, 10%, and 50% of predicted EHR-ED probability. To evaluate predictive performance, an optimal PRS threshold was determined using the ROC curve and Youden's $J$ statistic, classifying individuals into risk groups. We then calculated the sensitivity and specificity of the predicted versus observed ED status.

## Phenome-Wide Association Study (PheWAS)

We conducted a pheWAS of rs78677597, the lead SNP (at $p = 5.35 \times 10^{-139}$) in our EUR GWAS, to examine its association with traits besides EHR-ED. We used this approach because this lead SNP exceeds the next lead SNP in significance by 100 orders of magnitude. We used the Finngen, UKBB, and MVP meta-analysis (https://mvp-ukbb.finngen.fi), which was released in November 2024 and includes 330 phecodes, based mainly on EUR participants. After Bonferroni correction for 330 tests (0.05/300) the significance threshold was set $p = 1.67 \times 10^{-4}$.

## Transcriptome-Wide Association Study (TWAS)

We conducted TWAS using GTEx_v8[137], which provides expression data of 49 tissues in EUR samples. We used the 1000 Genomes dataset as an LD reference. Using FUSION[138], we identified associated genes, then processed the results to distinguish conditionally independent genes. In 49 tissues, there were a total of 300,187 genes measured (an average of $6126 \pm 2787$ per tissue). We therefore used a Bonferroni correction for 300,187 tests (0.05/300,187) to set a $p$-value threshold of $1.66 \times 10^{-7}$.

## Summary-based Mendelian randomization (SMR)

We conducted SMR using GTEx_v8[137] implemented in the SMR portal[139] to examine the causal effect of gene expression in various tissues on EHR-ED. We included in this analysis tissues that had at least one significant effect in the TWAS analysis, as well as the prostate and the entire set of brain regions available in the database. For each tissue, we tested whether gene expression had a causal effect on EHR-ED. Heterogeneity In Dependent Instruments (HEIDI) test was conducted to assess whether the variants are causal for both the exposure and target traits. Across all genes and tissues included in the analysis, we conducted 59,287 tests. After Bonferroni correction, $p$-value threshold was set $8.43 \times 10^{-7}$.

## Drug repurposing

We used the drug.MATADOR database, implemented in ShinyGo 0.82[140], to identify potential drug–target interactions, using 67 genes that were genome-wide significant (GWS) and/or had a significant effect in the gene-based analysis (MAGMA) and/or the TWAS (FUSION) in EUR, AFR, and cross-ancestry analyses.

## Genomic Structural Equation Modeling (gSEM)

gSEM[141] was performed on EHR-ED and 12 traits of interest that had significant genetic correlations with EHR-ED. Exploratory factor analysis (EFA) was performed on odd chromosomes, followed by confirmatory factor analysis (CFA) on even chromosomes to provide "independent" sets of chromosomes to identify and then confirm the factor structure. EFA models containing 1–8 factors were evaluated based upon eigenvalues, sum of squared (SS) loadings, cumulative variance explained, and the distribution of variance explained across the respective factors.

## Reporting summary

Further information on research design is available in the Nature Portfolio Reporting Summary linked to this article.

## Data availability

Supplementary Figs. 1–5 and Supplementary Tables 1–8 are in the supplementary information. Supplementary Data 1–13 are in the supplementary data. GWAS summary statistics generated in this study are available at the Gelernter Lab website (https://medicine.yale.edu/lab/gelernter/stats/) and also in the figshare platform (https://doi.org/10.6084/m9.figshare.30505799).

## Code availability

This study adapted and reused code from previously published works[111,114,115,135,136,138,139,141] from github (https://github.com/bulik/ldsc, https://github.com/chrchang/plink-ng, https://github.com/statgen/METAL, https://github.com/n-mounier/MRlap, https://github.com/josefin-werme/LAVA, https://github.com/GenomicSEM/GenomicSEM, https://github.com/gusevlab/fusion_twas, https://github.com/jianyanglab/SMR-Portal). The reused and adapted code complies with the original license terms (MIT, GPLv2, GPLv3). All code adaptations credit the original authors as required by the license. Users are encouraged to review the original publication for further context on the reused code's functionality.

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

## Acknowledgements

We acknowledge All of Us participants for their contributions, without whom this research would not have been possible. We also thank the National Institutes of Health's All of Us Research Program for making available the participant data examined in this study (researchallofus.org). Genotype-tissue expression data used for the analyses described in this manuscript were obtained from the GTEx Portal on 04/25/2025. This research was supported by NIH grants R01DA058862 (J.G.), R01DA054869 (J.G.), and K01DA058807 (J.D.D.) and by funding from the Department of Veterans Affairs Office of Research and Development, USVA grant I01CX001849 (JG), and a Career Development Award CDA-2 from the Veterans Affairs Office of Research and Development (DFL; IK2BX005058).

## Author contributions

U.B. conceived the study, conducted data analysis, and drafted the article. Y.C. and J.D.D. conducted data analysis. H.Z. and D.F.L. provided scientific consultancy and guidance. J.G. conceived the study, revised the article, and supervised the work. All coauthors reviewed the article.

## Competing interests

J.G. is paid for editorial work by the journal *Complex Psychiatry*. The remaining authors declare no competing interests.
