## [Transparent Peer Review file · Nature Communications]

Multi-Ancestry Investigation of the Genomics of Erectile Dysfunction

Corresponding Author: Professor Joel Gelernter

Version 0:

Reviewer comments:

Reviewer #1

(Remarks to the Author)

This manuscript will be a great addition to the knowledge surrounding ED genetics and discovers novel loci via meta-analysis of the existing studies with their new All of US analysis.

1. In the abstract and elsewhere when introducing the SIM1 variant, it should be more obviously stated that this was previously discovered in the same introductory sentences.
2. There are 40 lead snps from this GWAS which significantly expands the catalog of known ED SNPs.
3. The Introductory text should have more details (Size, where from, ancestry mix, etc.) on the existing GWAS that are meta-analyzed. They are mainly just cited but little is described until much later in the Methods. This information should not be relegated to the supplemental materials.
4. Table 1 is included as a supplemental material. This should be in the main text and include all the cohorts looked at.
5. The author's are reporting SNP wise association with a presumed gene name like this:
PHF21B*rs8141413
Being near a gene does not indicate that it is the causal gene and should not confuse readers by making this annotation here. The authors should report SNP rsID and a chromosome, position and effect size (OR) with confidence intervals in the text and in the summary table. Assumptions about the nearby genes should be left to later interpretation based on the MR and eQTL/pQTL annotations that can be added elsewhere. Saying the SNPs map to coding and noncoding genes is not correct. They are near or in the genes as loci. They only map to coding if they change the protein translated in the CDS as in an exon. These should be referred to as independent loci and not to as genes.
6. For all the reported GWS snps, how many of the minor alleles are protective versus increased risk? The snpwise summary stats for the largest main meta-analysis should be included as a table in the main text and not just in the supplement.
7. For the Polygenic score results. The authors should produce a deciles plot of the predicted probability (x-axis) which has the true prevalence plotted on the Y-axis.
A study of the extremes of the predicted ED probability using a relative risk model comparing the top 2.5%, 5% and 10% of the predicted ED probability with the middle 2 quartiles of the prediction probability.
They should also produce a cut point of the PRS for the optimal sensitivity and specificity with the predicted and known ED in a 2x2 table and report the AUC of the prediction.
8. MR analysis should also include pQTLs (Ferkinstad 2021) and/or eQTLs (GTEx) to show the causality.
9. Endometriosis is included in the gSEM. This is a trait of women, who do not have ED. While it is interesting that endometriosis has some genetic architecture that is in common with ED, I am not sure it is appropriate to include in this model. Please elaborate. A model should be run without this as well perhaps?
10. Make of Table in the main text that details the novel versus known SNPs with reference to the GWAS where they were discovered. The supplement should have a more detailed table of this information.

Reviewer #2

(Remarks to the Author)

General:

The topic is important and the authors leverage an important new resource, All of Us, to advance the field. The metaanalysis

and exploration of genomic architecture build on earlier work, are well planned and executed, and represent important new discoveries and further replication of the main genetic locus associated with male ED.

Title:

The term "reported ED" does not follow convention. Erectile dysfunction is a condition. It can be characterized as self-reported or administrative. It can be defined by electronic health records. But the term "reported ED" is meaningless to the average reader; the authors would be advised to leave the term out of the title.

Abstract:

The abstract is weak because it fails to include some key issues. The statement about the primacy of arteriogenic causes is probably an oversimplification reflecting the quality of the references used to introduce the topic. Age is strongly associated with ED and nowhere in the entire manuscript do the authors make any allusion to this or to try to account for it in this cross sectional analysis. The abstract should include the number of participants in the AoU GWAS, because that is the primary focus of this paper, which drives the integration with the other data sets in the meta-analysis. The abstracts should report the GWAS significant variant and possibly the other genes found with specific levels of significance.

Background:

The introduction to ED is probably overly broad and makes some linkages such as psychogenic ED and depression related ED that do not align with what most clinicians (and ICD-10) would consider as non-organic and organic ED. The mention of obesity fails to capture the notion that obesity is a significant health problem that is strongly associated with organic ED regardless of genetic issues. Lastly, the concept of self-report as this reviewer conceives it is related to definitions that do not require some independent physical assessment of ED. In a way, all ED is self reported. When a physician and a patient are discussing ED, it might be that the patient has a chief complaint of ED, or it might be mention of symptoms, or it might be elicited. That said, all three are legitimate examples of ED.

Methods:

The methods are under developed in terms of the ED phenotyping. This reviewer is concerned about effort to develop their methods beyond administrative definitions. They do not appear to set any minimum requirements for the number of times there was an ICD10 diagnosis code for ED. Nor to acknowledge that the information about PDE5 -I prescriptions, including frequency type and healthcare setting is quite underdeveloped. Because MVP and other biobanks include health maintenance organizations, pharmacy data may be very accurate. In AoU, they provide no information about the likely heterogeneity of payor coverage of PDE5-I, thus potentially introducing bias based on phenotyping.

No exclusion criteria are mentioned other than pulmonary hypertension. Did the authors consider that prostate cancer treatments of any kind, or other forms of radical pelvic surgery, might lead to bias in the analyses?

With regard to phenotyping in the meta-analysis, the authors don't indicate what phenotyping was used in MVP. Was this again ICD 10 codes? The VA contains baseline survey data, which lists a number of self-reported conditions. Did the authors consider this? The EHRED definition from reference 17 is not readily accessible to this reviewer, despite going in and trying to read that paper in Science.

There is no mention of phenotype validation. Was there an effort for manual chart validation or other assessment of the accuracy of the EHRED phenotype? Did the authors try to harmonize the definitions across the different biobanks used in the metanalysis?

It's also unclear why the authors did not include the data from reference 15 in the metaanalysis, which has by far the most ED cases of any of the biobanks and has robust information on severity in addition.

Regarding the polygenic score, was there any attempt to validate the score in one of the other cohorts? It certainly appears to have sample size to take a more in-depth approach rather than the first pass shown here.

Results:

The authors provide no information on the demographics of the cohorts such as Age, BMI or other medical comorbidity for their primary GWAS analysis.

It's of note that there are about 20,000 cases of ED out of a total of 120,000 men in AoU. In the KP GERA cohort in Ref. 15, the numbers were substantially closer to the expected prevalence of ED in the United States. The numbers suggests that either the cohort is much younger than others cohorts, or there is vast degrees of underreporting. A comment on this would be appropriate.

The novel association of ED with ESR1 is important. It would be interesting to expand on this in light of prior work showing GWAS significant association with the FSHR with ED in men of AFR ancestry undergoing radiotherapy for prostate cancer. See Kerns SL, Ostrer H, Stock R, Li W, Moore J, Pearlman A, Campbell C, Shao Y, Stone N, Kusnetz L, Rosenstein BS. Genome-wide association study to identify single nucleotide polymorphisms (SNPs) associated with the development of erectile dysfunction in African-American men after radiotherapy for prostate cancer. *Int J Radiat Oncol Biol Phys.* 2010 Dec 1;78(5):1292-300. doi: 10.1016/j.ijrobp.2010.07.036. PMID: 20932654; PMCID: PMC2991431.

Discussion:

The reviewer is surprised that the discussion begins with metaanalysis without mention of the GWAS in AoU at the start. It's

as if they're jumping to the other information before focusing on the primary driver of their analysis.

It would've been helpful to involve a molecular scientist or other genetics expert to discuss the variant that was replicated and any thoughts they have about why RS17185536 versus 786-77597 show up in the various ancestry and other analyses. Particularly because the former has functional data that supported biological mechanism.

The discussion on Sim1 is very underdeveloped. The authors don't acknowledge an entire line of investigation on the role of the Sim1 transcription factor as a key regulator of the melanocortinergic pathway and this system's contribution to erectile functioning; this is a major oversight.

The shortcomings section acknowledged future opportunity to look at severity, but nothing about the age of patients in the cohort, the potential use of age of onset analyses that might be related to ED.

The testis function discovery is interesting and appears straightforward; however, ED is not highly correlated with testicular problems except severe forms of hypogonadism. Expanding on these diagnoses and linking the PheWAS and GWAS data for the most significant hits would be useful.

The discussion of ESR1 is well-developed, but this reviewer is concerned that prostate cancer patients may be contributing to the cases in a overrepresented fashion, and introducing unrecognized bias

The mendelian randomization strikes this reviewer as missing important further depth. For example, the relationship between morbid obesity and erectile function, in terms of causality, suggest to this reviewer that there's something wrong with the model. To conclude that ED is a cause of morbid obesity must miss some intervening intermediate factors.

References:

Up to date

Figures:

The reviewer found it frustrating that there were no captions for the supplemental figures making it hard to understand them within the context of the paper.

Version 1:

Reviewer comments:

Reviewer #1

(Remarks to the Author)

Great revision.

Table 2 might be a little large for the print version.

Perhaps a condensed version of just the counts of novel/known snps for the main text and then the larger version can be put in the supplement.

Accept for publication.

Reviewer #2

(Remarks to the Author)

The authors have been responsive and the revised manuscript provides important details in the methods section and in the discussion.

The fact that the cohort has such high representation of prostate cancer amongst the cases vs. controls could lead to confounding. Are some of the genes actually prostate cancer genes? Did the authors do sensitivity analyses removing patients with prostate cancer from the cohort?

REVIEWERS COMMENTS

Reviewer #1 (Remarks to the Author):

This manuscript will be a great addition to the knowledge surrounding ED genetics and discovers novel loci via meta-analysis of the existing studies with their new All of US analysis.

1. In the abstract and elsewhere when introducing the SIM1 variant, it should be more obviously stated that this was previously discovered in the same introductory sentences.

Thank you for the positive comments. We added the following sentence to the abstract (page 2):

“This region was previously associated with ED in prior GWAS.”

2. There are 40 lead snps from this GWAS which significantly expands the catalog of known ED SNPs. The Introductory text should have more details (Size, where from, ancestry mix, etc.) on the existing GWAS that are meta-analyzed. They are mainly just cited but little is described until much later in the Methods. This information should not be relegated to the supplemental materials.

We agree that this information is important and have now provided a more prominent place in the manuscript. We elaborated in the introductory part about the genetic ancestries and sample size of the meta-analyzed GWAS (page 6):

*“...then meta-analyzed our results with previously published ED GWAS data, **which include a total of 824,472 subjects of EUR ancestry and 94,867 individuals of AFR ancestry**”*

We also expanded the methods section discussing the cohort, elaborating regarding the number of cases included in each ancestry (page 7):

*“...gained through the following cohorts: UK Biobank (UKBB), the Estonian Genome Center of the University of Tartu cohorts, hospital-recruited Partners HealthCare Biobank (PHB) cohort **(6,175 EUR cases)** [21], the Million Veteran Program (MVP) **(110,823 EUR and 47,384 AFR cases)** [22] and FinnGen **(2,886 EUR cases)** [32].”*

3. Table 1 is included as a supplemental material. This should be in the main text and include all the cohorts looked at.

We expanded the table, adding data regarding the various cohorts which are included in each of the GWAS incorporated in the meta-analysis. We moved the table to the results section in the main manuscript: **Table 1: Sample size across the various cohorts** (we left the heritability

estimates in the supplementary files, as **table S1: Heritability estimates across the various EUR cohorts**).

4. The author's are reporting SNP wise association with a presumed gene name like this:

PHF21B*rs8141413

Being near a gene does not indicate that it is the causal gene and should not confuse readers by making this annotation here. The authors should report SNP rsID and a chromosome, position and effect size (OR) with confidence intervals in the text and in the summary table. Assumptions about the nearby genes should be left to later interpretation based on the MR and eQTL/pQTL annotations that can be added elsewhere. Saying the SNPs map to coding and noncoding genes is not correct. They are near or in the genes as loci. They only map to coding if they change the protein translated in the CDS as in an exon. These should be referred to as independent loci and not to as genes.

We revised the GWAS results section. We changed *PHF21B**rs8141413 to:

"rs8141413 (intronic to PHF21B; p=4.18x10⁻¹⁹)" (page 12)

We have done the same regarding all similar instances. Because our meta-analysis was n_{eff} -weighted, we do not have OR, but only Z scores (this is the output the METAL provides for this kind of analysis) – and we present them all in table 2, along with chromosome and position, as is now stated in the text:

*"full details regarding chromosome, position and Z scores are presented in **Table 2**".* (page 12)

We also changed the following paragraph in the discussion (in **bold** letters):

*"In total, we found 40 lead SNPs in EUR, mapping **close to or within** 19 coding and non-coding genes, and two lead SNPs in AFR, **of which one is intronic to a coding gene**. In the cross-ancestral analysis, we found 51 lead SNPs, **mapping close to or within** a total of 26 different coding and non-coding genes. PRS analysis in EUR revealed that our GWAS explains 9.2% of the variance in EHR-ED."* (page 28)

5. For all the reported GWS snps, how many of the minor alleles are protective versus increased risk? The snpwise summary stats for the largest main meta-analysis should be included as a table in the main text and not just in the supplement.

We moved table S2 to the results section of the main text, as **Table 2: Lead SNPs**. In addition, we reported the number of protective SNPs in each of the meta-analyses:

*“In the EUR meta-analysis, we identified a total of 40 lead SNPs in 27 genomic risk loci, 10 are novel, **and for 17 the minor allele is protective**”* (page 12)

*“In AFR, we found two statistically significant variants: rs17185536-T ($p=1.17 \times 10^{-9}$) and rs55659406 (intronic to RABGAP1L; $p=4.79 \times 10^{-8}$), the latter is novel and **for both the minor allele increases risk of EHR-ED.**”* (page 12)

*“Cross-ancestry (EUR-AFR) meta-analysis revealed an increase from 40 lead SNPs in EUR and 2 in AFR to 51 lead SNPs (in 34 genomic risk loci, **for 24 the minor alleles are protective**)...”* (page 12-13)

6. For the Polygenic score results. The authors should produce a deciles plot of the predicted probability (x-axis) which has the true prevalence plotted on the Y-axis. A study of the extremes of the predicted ED probability using a relative risk model comparing the top 2.5%, 5% and 10% of the predicted ED probability with the middle 2 quartiles of the prediction probability.

They should also produce a cut point of the PRS for the optimal sensitivity and specificity with the predicted and known ED in a 2x2 table and report the AUC of the prediction.

We thank the reviewer for these suggestions, which add an important value for the PRS analysis. We expanded the analyses as suggested, with these additions in the text and the referred supplementary figure and table:

Methods:

“EHR-ED odds were tested in individuals in the top 2.5%, 5%, 10% and 50% of predicted EHR-ED probability. To evaluate predictive performance, an optimal PRS threshold was determined using the ROC curve and Youden’s J statistic, classifying individuals to risk groups. We then calculated sensitivity and specificity of the predicted versus observed ED status.” (page 10)

Results:

*“After Bonferroni correction (p -value threshold=0.0023), in EUR we found that EHR-ED (meta-analysis excluding AoU) PRS is significantly associated with EHR-ED in the AoU sample, in all but two of the 11 examined SNP inclusion p -value thresholds (the ones without significant association were $p=0.01$ and $p=0.0001$). The strongest effect was achieved using a SNP inclusion cutoff of $p=0.5$, explaining 9.2% of the phenotypic variance ($p=2.20 \times 10^{-36}$) (Table S4). Nevertheless, the predictive value of this model was limited (AUC = 0.52), in accordance with positive modest association between EHR-ED PRS (under a p -value threshold of 0.5) and EHR-ED risk (**Figure S5**). Individuals in the top 2.5%, 5% and 10% of predicted EHR-ED probability had 53% (OR=1.53 ± 0.14), 18% (OR = 1.18 ± 0.13) and 16% (OR = 1.16 ± 0.08) higher odds of EHR-ED compared to individuals in the middle 50% of predicted probabilities. ROC analysis identified an optimal probability cutoff for sensitivity and specificity for SNP inclusion in PRS at*

$p=9.23 \times 10^{-6}$. Using this threshold, 4,697 cases were correctly predicted (true positives) and 17,583 controls were incorrectly predicted as high risk (false positives), yielding a sensitivity of 0.28 and specificity of 0.75 (**Table S5**). (page 18-19)

7. MR analysis should also include pQTLs (Ferkinstad 2021) and/or eQTLs (GTEx) to show the causality.

We added summary-based MR (SMR) analysis to assess the causal effects of EHR-ED on gene expression (eQTL) using the SMR-portal. Results are presented in tables S15. We added the following text in the manuscript.

Methods:

“Summary-based Mendelian Randomization (SMR)

We conducted SMR using GTEx_v8 [63] implemented in the SMR portal [65] to examine the causal effect of gene expression in various tissues on EHR-ED. We included in this analysis tissues that had at least one significant effect in the TWAS analysis, as well as prostate and the entire set of brain regions available in the database. For each tissue, we tested whether gene expression had a causal effect on EHR-ED. Heterogeneity In dependent instruments (HEIDI) test was conducted to assess whether the variants are causal for both the exposure and target traits. Across all genes and tissues included in the analysis, we conducted 59,287 tests. After Bonferroni correction, p-value threshold was set to 8.43×10^{-7} .” (page 11)

Results:

“SMR

Two genes had a significant effect in the entire brain: CTNNB1 had a negative effect ($\beta=-0.091$, $p=1.79 \times 10^{-7}$), suggesting that the genetic variants associated with brain expression mapping to this gene are negatively associated with EHR-ED; C1GALT1 had a positive effect ($\beta=0.051$, $p=3.63 \times 10^{-7}$), suggesting that the genetic variants associated with brain expression mapping to this gene are positively associated with EHR-ED (Table S16).” (page 26)

Discussion:

“CTNNB1, which encodes β -catenin, was one of the most strongly associated genes with EHR-ED in the gene-based analysis; it was associated with low enrichment in the amygdala (in the TWAS) and in the entire brain (in the SMR) of EHR-ED subjects, suggesting that β -catenin downregulation in the brain may increase ED risk.” (page 33-34)

8. Endometriosis is included in the gSEM. This is a trait of women, who do not have ED. While it is interesting that endometriosis has some genetic architecture that is in common with ED, I am not sure it is appropriate to include in this model. Please elaborate. A model should be run without this as well perhaps?

Including a female-specific trait in our analyses indeed deserves a better explanation, which we now added in the discussion:

“Endometriosis, on the other hand, is a female-specific trait that is also affected by ESR1 [32], and is associated with sexual dysfunction [117]. In consideration of its somewhat congruent relationship to sexual dysfunction in men, we sought to discover whether this trait is genetically correlated with EHR-ED, and found a significant genetic correlation between the traits, suggesting shared genetic architecture between these sex-specific traits. Nevertheless, these traits did not load under the same factor in the gSEM analysis, in which endometriosis loaded alongside traits of a psychiatric nature, such as depression and PTSD.” (page 32-33)

We feel that endometriosis had an important place in the gSEM analysis, for its significant genetic correlation with ED, as well as its phenotypic (partial) resemblance to ED – even though these traits cannot ever appear together. For these reasons, and in order to avoid an increased burden of multiple testing that we would incur if we repeated a new set of gSEM analyses, we preferred not to exclude endometriosis from this analysis.

9. Make of Table in the main text that details the novel versus known SNPs with reference to the GWAS where they were discovered. The supplement should have a more detailed table of this information.

In the new **table 2** in the main text, we included a reference column, which refers to previous GWAS in which the SNPs were discovered. In cases when there is no available reference, it is stated that the SNP represents a novel hit. This suggestion from the reviewer helped us identify that only nine SNPs were novel – and not ten as was previously mentioned. That number was corrected in the results section:

“In the EUR meta-analysis, we identified a total of 40 lead SNPs in 27 genomic risk loci, nine are novel...” (page 12)

Reviewer #2 (Remarks to the Author):

General:

The topic is important and the authors leverage an important new resource, All of Us, to advance the field. The metaanalysis and exploration of genomic architecture build on earlier work, are well planned and executed, and represent important new discoveries and further replication of the main genetic locus associated with male ED.

Title:

1. The term “reported ED” does not follow convention. Erectile dysfunction is a condition. It can be characterized as self- reported or administrative. It can be defined by electronic health records. But the term “reported ED” is meaningless to the average reader; the authors would be advised to leave the term out of the title.

We changed the title, as suggested by the reviewer, to: “Multi-ancestry Investigation of the Genomics of Erectile Dysfunction” (page 1)

Abstract:

2. The abstract is weak because it fails to include some key issues. The statement about the primacy of arteriogenic causes is probably an oversimplification reflecting the quality of the references used to introduce the topic. Age is strongly associated with ED and nowhere in the entire manuscript do the authors make any allusion to this or to try to account for it in this cross sectional analysis. The abstract should include the number of participants in the AoU GWAS, because that is the primary focus of this paper, which drives the integration with the other data sets in the meta-analysis. The abstracts should report the GWAS significant variant and possibly the other genes found with specific levels of significance.

We revised the abstract as advised by the reviewer. We emphasized the importance of age in ED, added the sample size in All of Us, and mentioned other notable GWS hits besides the chromosome 6 region (page 2):

*“Erectile dysfunction (ED) is **often** attributable to arterial insufficiency, but can also arise as a result of cardiovascular disease, drug use, psychological factors, and hormonal imbalance, or other biological issues, **such as diabetes and prostate cancer**”.*

“Its prevalence rises with age, with an average of 30% over the age of 40 compared to <10% in younger people”

“...in the All of Us database [N(EUR)=88,722 (16,983 cases), N(AFR)=30,448 (4,215 cases)], then meta-analyzed our findings...”

“Other notable genes include CASC19, PHF21B and ESR1....”

Background:

3. The introduction to ED is probably overly broad and makes some linkages such as psychogenic ED and depression related ED that do not align with what most clinicians (and ICD-10) would consider as non-organic and organic ED. The mention of obesity fails to capture the notion that obesity is a significant health problem that is strongly associated with organic ED regardless of genetic issues. Lastly, the concept of self-report as this reviewer conceives it is related to definitions that do not require some independent physical assessment of ED. In a way, all ED is self reported. When a physician and a patient are discussing ED, it might be that the patient has a chief complaint of ED, or it might be mention of symptoms, or it might be elicited. That said, all three are legitimate examples of ED.

Thank you for these important points that require clarification in the article. We expanded the introduction to include the suggested additions regarding psychogenic ED and obesity:

*“The most common direct cause of **organic ED** is reduced blood flow and arterial insufficiency, which may result from vascular disease, often in association with smoking and diabetes.*

Besides age, diabetes is the strongest risk factor for ED, and cardiovascular diseases are strongly implicated too [5, 6]. There is also high comorbidity between obesity and ED: up to 80% of the people who report ED are overweight or obese [7]. This association between cardiovascular traits, obesity and ED is strengthened by findings of improved sexual functioning due to lifestyle changes aimed at lowering cardiovascular risk [8].”

(page 4)

*“...Psychological factors can also play a major role, in psychogenic ED; **that is, ED that is predominantly caused by psychological factors such as symptoms of anxiety and stress without an organic explanation [1].”*** (page 4)

Regarding the issue of self-report: we agree with the reviewer that (nearly) all ED is self-reported, or to put it more accurately – essentially all ED that is documented in EHR is self-reported; therefore, ED that is not self-reported is not documented in EHR. That was the point we intended to make: there are people with ED who do not report it, and they may differ phenotypically (and therefore may also differ genetically) from those who do report the issue. For example, men with ED who do not wish to be sexually active, may not report that they have ED in a medical context. Nevertheless they still may have ED. In later parts of this manuscript, this point is important, and that is the reason we emphasized it in the introduction. As we wrote in the manuscript: “In this study we aimed to investigate the genetic architecture of ED. Our analysis was based on electronic health record (EHR) information, mostly based on self-report (albeit to a physician), and hence likely to be missing a portion of the population who do not report ED, even though they experience ED symptoms [2, 3]. Therefore, we defined our phenotype as EHR-ED, to differentiate our phenotype from population-based ED figures.”

Methods:

4. The methods are under developed in terms of the ED phenotyping. This reviewer is concerned about effort to develop their methods beyond administrative definitions. They do not appear to set any minimum requirements for the number of times there was an ICD10 diagnosis code for ED. Nor to acknowledge that the information about PDE5 -I prescriptions, including frequency type and healthcare setting is quite underdeveloped. Because MVP and other biobanks include health maintenance organizations, pharmacy data may be very accurate. In AoU, they provide no information about the likely heterogeneity of payor coverage of PDE5-I, thus potentially introducing bias based on phenotyping.

No exclusion criteria are mentioned other than pulmonary hypertension. Did the authors consider that prostate cancer treatments of any kind, or other forms of radical pelvic surgery, might lead to bias in the analyses?

There are some important points here that we now address. This study was originally designed to be a meta-analysis, thus we considered the nature of the other available sources when we planned our new analysis in All of Us, with the intention of minimizing phenotypic heterogeneity. While the MVP pheWAS (Verma et al) did define a necessary minimum of two ICD registrations of ED to define a subject as a case, the Bovijn et al analysis – also included in our meta – did not. Our decision to use the more-inclusive / less-stringent approach was done with these concerns in mind. The rg between the aforementioned studies was ~1, indicating the two phenotypic approaches led, in these cases, to similar genomic constructs. We added these important concerns in the limitations (as well as a few others, to address other concerns made by this reviewer in other comments):

“Third, we aimed to conduct our analysis in AoU as similarly as possible to the already-completed analyses in other cohorts included in the meta-analysis. For this reason, we did not include or exclude subjects according to the number of inpatient/outpatient visits (i.e., one visit was enough to include a subject as an ED case). Similarly, we did not include an age limit for the analysis, although ED is more common in older ages (we did, however, use age as covariate in the regression), nor did we exclude subjects with a history of prostate cancer treatment, which is a risk factor for ED [133]. Considering the relatively large proportion of prostate cancer history among ED subjects in our analysis (17.8%, compared to 5.3% in the control group in EUR; 16%, compared to 1.9% in the controls group in AFR), it is plausible that inclusion of prostate cancer subjects could affect the results. These inclusion criteria enabled us to meta-analyze phenotypically-similar traits, but at the potential cost of increased noise. Nevertheless, in terms of the strongest GWS locus, our main analysis achieved similar results to previous ED GWAS (with increased power), which either included [21, 22] or excluded [20] subjects with a history of prostate cancer, suggesting that our results are valid beyond intervening variables.” (page 37-38)

5. With regard to phenotyping in the meta-analysis, the authors don't indicate what phenotyping was used in MVP. Was this again ICD 10 codes? The VA contains baseline survey data, which lists a number of self-reported conditions. Did the authors consider this? The EHRED definition from reference 17 is not readily accessible to this reviewer, despite going in and trying to read that paper in Science.

This is a good point, which applies to other datasets too. We added this information in the 'Phenotype' column in the newly added **Table 1**. We also added this sentence in the Methods section (in **bold**) [note: Verma et al (previously ref.17) is now ref. 22]:

*"We then conducted three separate meta-analyses: within-EUR, within-AFR and cross-ancestry, including our results from AoU combined with previously published data, gained through the following cohorts: UK Biobank (UKBB), the Estonian Genome Center of the University of Tartu (EGCUT) cohorts, hospital-recruited Partners HealthCare Biobank (PHB) cohort (6,175 EUR cases) [21], the Million Veteran Program (MVP) (110,823 EUR and 47,384 AFR cases) [22] and FinnGen (2,886 EUR cases) [32]. **Cases in these cohorts were defined by ICD-10 code, drug prescription records and self report.**"* (page 7)

6. There is no mention of phenotype validation. Was there an effort for manual chart validation or other assessment of the accuracy of the EHRED phenotype? Did the authors try to harmonize the definitions across the different biobanks used in the metanalysis?

We do not have access to subjects' personal charts so manual chart validation was not possible. There are differences between the ways the ED phenotype was defined in the various cohorts, and we did our best to account for these differences when we constructed our phenotype in All of Us. We applied inclusion criteria similar to those applied by Bovijn et al (ref. 21), as described in our methods section:

"EHR-ED was defined according to (a) participants' electronic health record (EHR) (ICD-10 code N52) or (b) participants being prescribed with one or more of the ED drugs (PDE5-inhibitors) sildenafil (when prescribed as Viagra; this same medication is also used to treat pulmonary hypertension under a different brand name), tadalafil, and vardenafil. Subjects who met at least one of these criteria were included in the analysis as cases." (page 6d)

Inclusion criteria for the MVP study (Verma et al; ref. 22) and FinnGen (ref. 32) were based on ICD codes too. Please see also comments on the previous page regarding our additions in the limitations parts and in table 1.

7. It's also unclear why the authors did not include the data from reference 15 in the metaanalysis, which has by far the most ED cases of any of the biobanks and has robust information on severity in addition.

[note: Jorgenson et al (previously ref.15) is now ref. 20]

We hoped to address this comment by adding data published in ref.20. While doing so would very likely have increased our power for locus discovery, we also note that although ref.20 is an excellent paper and presented important progress in the field, by the time we assembled data for our meta-analysis it was no longer the largest – the largest sample was that presented in the MVP SCIENCE paper, ref. 22, which had 158,207 ED cases (and 319,169 controls) across ancestries, compared to 14,215 cases (and 22,434 controls) in ref. 20. The increment for our study from ref 20 data would have been approximately 7.5% of the total number of cases included in our analysis already; that is, a modest increase. We approached the authors of that paper, and were asked by them to include three co-authors from their group in exchange for providing us with the data from their article published in 2018. Three new authorships for our paper with a current total of six co-authors, in return for access to data published seven years ago that would increase our sample size modestly, seemed disproportionate. We therefore did not pursue these data further.

8. Regarding the polygenic score, was there any attempt to validate the score in one of the other cohorts? It certainly appears to have sample size to take a more in-depth approach rather than the first pass shown here.

We currently do not have access to other cohorts with ED data (besides MVP, in which we cannot analyze individual-level ED data as our authorizations do not include any use of data regarding this trait).

Results:

9. The authors provide no information on the demographics of the cohorts such as Age, BMI or other medical comorbidity for their primary GWAS analysis.

Thank you for this suggestion, which was indeed missing from our manuscript, and add further depth for understanding the phenotype. We added a supplementary table (**table S1**) with details of age, BMI and history of prostate cancer and type 2 diabetes in the EUR and AFR All of Us subjects. We also included this additional text in the first paragraph of the results section:

“Cases were on average 10 years older than controls; prevalence of prostate cancer was 3.3 times higher in ED EUR subjects (compared to controls) and 8.4 times higher in AFR; type 2 diabetes (T2D) prevalence was twice as high in ED EUR subjects and four times higher in AFR (Table S1).” (page 12)

We also added the following in the discussion (in **bold**):

*“...consistent with a known genetic and phenotypic association between ED and T2D [3, 21], **which was also seen in our data (see Table S1).**”* (page 33)

*“...prostate cancer [57, 92-94], a trait we found to have a bidirectional causal relationship with EHR-ED, in accordance with known strong comorbidity between the two traits [95, 96] **and with increased prevalence of prostate cancer among ED subjects in our sample as well. . This association is likely causal, as both prostate cancer and prostate cancer treatments may lead to ED in patients [95, 96].**”* (page 31)

10. It's of note that there are about 20,000 cases of ED out of a total of 120,000 men in AoU. In the KP GERA cohort in Ref. 15, the numbers were substantially closer to the expected prevalence of ED in the United States. The numbers suggests that either the cohort is much younger than others cohorts, or there is vast degrees of underreporting. A comment on this would be appropriate.

We added a brief discussion of the diverse case/control and age distribution in the limitations:

“Fourth, case/control distribution in our samples vary widely, which may be associated with age distribution: in MVP, a relatively older sample (mean age 62) [22], the prevalence of ED was 28.9%, compared to 19.1% in the younger cohort of AoU (mean age 58). Nevertheless, in Finngen only 1.32% of the subjects were ED cases, even though the mean age in this cohort (61) is closer to that of MVP [32], suggesting that other factors, such as diagnosis method, are of importance as well. These differences would have increased between-sample heterogeneity.” (page 38)

11. The novel association of ED with ESR1 is important. It would be interesting to expand on this in light of prior work showing GWAS significant association with the FSHR with ED in men of AFR ancestry undergoing radiotherapy for prostate cancer. See Kerns SL, Ostrer H, Stock R, Li W, Moore J, Pearlman A, Campbell C, Shao Y, Stone N, Kusnetz L, Rosenstein BS. Genome-wide association study to identify single nucleotide polymorphisms (SNPs) associated with the development of erectile dysfunction in African-American men after radiotherapy for prostate cancer. *Int J Radiat Oncol Biol Phys.* 2010 Dec 1;78(5):1292-300. doi: 10.1016/j.ijrobp.2010.07.036. PMID:

20932654; PMCID: PMC2991431.

We included this reference in the discussion, at the part discussing the ESR1 result:

*“In a study in healthy elderly men, raloxifene increased luteinizing hormone (LH), follicle-stimulating hormone (FSH) and sex steroid hormones, with no negative effects on erectile function [109]. **The gene encoding the FSH receptor was previously associated with ED following radiotherapy for prostate cancer in African Americans [110].**”* (page 32)

Discussion:

12. The reviewer is surprised that the discussion begins with metaanalysis without mention of the GWAS in AoU at the start. It's as if they're jumping to the other information before focusing on the primary driver of their analysis.

We did this to start the Discussion with what we considered the most important results, but have now added a discussion of the results of the AoU analysis at the beginning of the discussion:

*“ED is an important clinical problem that affects quality-of-life and reproductive capacity. Little is known about the genetic factors that affect risk for this disorder. **In our analysis in the AoU sample, we found one GWS locus in EUR, rs17185536, which is located in a non-coding region on chromosome 6. This result replicates several previous ED GWAS [20-22]. In AFR, in AoU we did not find any significant SNPs**”.* (page 28)

13. It would've been helpful to involve a molecular scientist or other genetics expert to discuss the variant that was replicated and any thoughts they have about why RS17185536 versus 78677597 show up in the various ancestry and other analyses. Particularly because the former has functional data that supported biological mechanism.

We expanded the discussion regarding the lead variants:

“The most robust effects were of rs78677597 in EUR ($p=5.35 \times 10^{-139}$) and of rs17185536 in AFR ($p=1.17 \times 10^{-9}$) and cross-ancestry ($p=5.30 \times 10^{-138}$), replicating previous findings of a strong association between this region on chromosome 6 and ED [20-22] (rs78677597 and rs17185536 are located in the same LD region). The rs78677597 variant is located within the non-coding region LOC105377911, and has been implicated as an inhibitor of SIM1, a gene located ~230 kb apart from this variant on chromosome 6. Data obtained from GTEx portal (gtexportal.org) indicated that rs17185536, mapping 11 kb from rs78677597 and with very high LD ($r^2=0.97$), is an eQTL for SIM1, with a negative effect on its expression. rs17185536 was previously identified as the lead SNP in a study of ED [20] and potentially the functional variant

in this region, affecting SIM1 expression [20, 68]. The fact that different lead SNPs were identified across ancestries likely reflects differences in LD patterns. The study that identified rs17185536 as the lead SNP was cross-ancestral [20], which aligns with the fact that the same SNP was the lead variant in our cross-ancestry meta-analysis too.” (page 28-29)

14. The discussion on Sim1 is very underdeveloped. The authors don't acknowledge an entire line of investigation on the role of the Sim1 transcription factor as a key regulator of the melanocortinergetic pathway and this system's contribution to erectile functioning; this is a major oversight.

We appreciate the suggestion to improve the discussion of the relevant biology, and expanded our discussion of *SIM1* to include this paragraph about its association to melanocortin-mediated changes in sexual function:

“SIM1 is also implicated in the effect of melanocortinergetic pathways on sexual function and behavior (melanocortin is an important mediator of sexual function, and specifically penile erection [72]): impaired sexual behavior in mc4r (melanocortin receptor 4)-knockout male and female mice was normalized by expressing mc4r exclusively on sim1 neurons [73,74]” (page 29)

15. The shortcomings section acknowledged future opportunity to look at severity, but nothing about the age of patients in the cohort, the potential use of age of onset analysis that might be related to ED.

Please see our answer to comment 10 (same review).

16. The testis function discovery is interesting and appears straightforward; however, ED is not highly correlated with testicular problems except severe forms of hypogonadism. Expanding on these diagnoses and linking the PheWAS and GWAS data for the most significant hits would be useful.

Addressing this comment allows us to provide additional ties between our findings and pathophysiology; we expanded our discussion of the *TEX41* finding (in **bold** letters):

*“In the TWAS, TEX41 was found to be in low enrichment in the testis, providing further indication of its possible role in sexual functioning. **This finding finds context in literature regarding the role played by TEX genes in male fertility, but not in sexual functioning [91]. Normal-range testicular function is generally not associated with ED.**” (page 30-31)*

17. The discussion of ESR1 is well-developed, but this reviewer is concerned that prostate cancer patients may be contributing to the cases in a overrepresented fashion, and introducing unrecognized bias

This concern deserves discussion, which we added in the manuscript:

“The role played by ER- α in prostate cancer [103], alongside the high comorbidity seen between prostate cancer and ED [104], raised the concern that the effect seen in ESR1 is driven by prostate cancer and not directly by EHR-ED; however, previous GWAS publications indicate that the effect of ESR1 on prostate cancer is low [105] or null [22, 32].” (page 31-32)

In addition, please see comment 4 (reviewer 2).

18. The mendelian randomization strikes this reviewer as missing important further depth. For example, the relationship between morbid obesity and erectile function, in terms of causality, suggest to this reviewer that there's something wrong with the model. To conclude that ED is a cause of morbid obesity must miss some intervening intermediate factors.

Indeed, it is unlikely that ED causes obesity. This is a valuable point about MR that needed to be clarified better in the manuscript. These findings represent genotypic causality, not phenotypic causality. We clarified this point in the text (in **bold** letters):

*“In the MR analysis we found that **having genetic variants which are associated with EHR-ED is a much stronger causal factor for morbid obesity than having genetic variants associated with morbid obesity is causal for EHR-ED**; obesity (not necessarily morbid) was found to be caused by **genetic liability to EHR-ED** but negatively correlated to, i.e. associated with reduction of, EHR-ED. **These findings indicate the important role of variants associated with EHR-ED in obesity and morbid obesity. These results do not mean that ED causes obesity per se (i.e., MR tells us about genetic, not phenotypic, causality).**”* (page 30)

19. References:

Up to date

20. Figures:

The reviewer found it frustrating that there were no captions for the supplemental figures making it hard to understand them within the context of the paper.

We now provided a separate file with all the supplementary figure captions.

Reviewer #2 (Remarks to the Author):

The authors have been responsive and the revised manuscript provides important details in the methods section and in the discussion.

The fact that the cohort has such high representation of prostate cancer amongst the cases vs. controls could lead to confounding. Are some of the genes actually prostate cancer genes? Did the authors do sensitivity analyses removing patients with prostate cancer from the cohort?

We expanded the limitations section to better acknowledge the issue (in **bold**):

*“...nor did we exclude subjects with a history of prostate cancer treatment, which is a risk factor for ED [104]. Considering the relatively large proportion of prostate cancer history among ED subjects in our analysis (17.8%, compared to 5.3% in the control group in EUR; 16%, compared to 1.9% in the controls group in AFR), it is plausible that inclusion of prostate cancer subjects could affect the results; **for example, some of the genes we identified in this study, ESR1 and CASC19, are genome-wide significantly associated with prostate cancer too [58-61, 70, 72], and therefore our results may be driven in part by these associations.** These inclusion criteria enabled us to meta-analyze phenotypically-similar traits, but at the potential cost of increased noise. Nevertheless, in terms of the strongest GWS locus, our main analysis achieved similar results to previous ED GWAS (with increased power), which either included [21, 22] or excluded [20] subjects with a history of prostate cancer, suggesting that our results are valid beyond intervening variables.”*

These issues were also previously addressed earlier in the discussion (page 32):

“The role played by ER- α in prostate cancer [103], alongside the high comorbidity seen between prostate cancer and ED [104], raised the concern that the effect seen in ESR1 is driven by prostate cancer and not directly by EHR-ED; however, previous GWAS publications indicate that the effect of ESR1 on prostate cancer is low [105] or null [22, 32].”

We also expanded our discussion of limitations to address the exploratory nature of the genetic correlation analysis with endometriosis (in **bold**):

*“Fifth, genetic correlations were mostly calculated vs traits that were measured in samples of both males and females, because male-only data are often not available and if available, less powerful. **This does not apply to the genetic correlation analysis with endometriosis, which is a female-only trait. Therefore, this is an analysis of a more exploratory nature, and the genetic correlations found between these traits cannot be directly asserted phenotypically.**”*